# Asymmetric Contrastive Objectives for Efficient Phenotypic Screening

Luke Nightingale[1]   Joseph Tuersley[1]   Scott Warchal[1]   Andrea Cairoli[1]   Jacob Howes[1]   Cameron Shand[1]
Andrew J. Powell[1 2]   Darren V.S. Green[2]   Amy Strange[1]   Michael Howell[1]

## Abstract

Phenotypic screening experiments produce many microscope images of cells under diverse perturbations, with biologically significant responses often subtle or difficult to identify visually. A central challenge is to extract image representations that distinguish activity from controls and group phenotypically similar perturbations. In this work we propose new adaptations of contrastive loss functions that incorporate experimental metadata as learned class vectors, and a geometrically inspired variant, called SPC, where class vectors are confined to the unit sphere and updated only by attractive terms (allowing more overlap of phenotypically similar classes). The approach is tested on two popular benchmarking datasets, BBBC021 and RxRx3-core; and we also evaluate performance on uncurated screens of HaCaT cells to gauge effectiveness in a realistic use-case scenario. We find we outperform prior methods across the three datasets and on a wide array of metrics measuring phenotype grouping, biological recall, drug-target interaction and mechanism-of-action inference. We also show we maintain this improved performance compared to models over 10x larger in parameter count, and that SPC can be used as an effective fine-tuning technique. The method is easy to implement and is well suited to settings with limited data or compute resources.

## 1. Introduction

High-content screening (HCS) is a high-throughput approach to drug discovery (Bickle, 2010; Bray et al., 2016). To facilitate testing many biological samples in parallel, experiments are performed with arrayed multi-well plates,

[1]The Francis Crick Institute, United Kingdom [2]GlaxoSmithKline, United Kingdom. Correspondence to: Luke Nightingale <luke.nightingale@crick.ac.uk>.

*Proceedings of the 43${}^{rd}$ International Conference on Machine Learning*, Seoul, South Korea. PMLR 306, 2026. Copyright 2026 by the author(s).

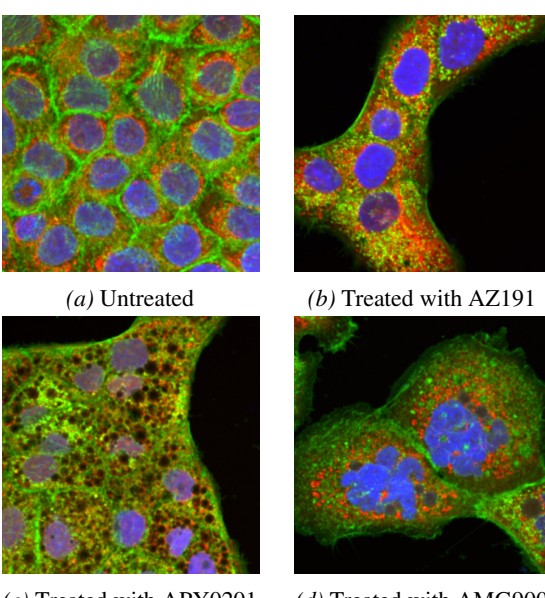

*(a)* Untreated          *(b)* Treated with AZ191

*(c)* Treated with APY0201     *(d)* Treated with AMG900

*Figure 1.* HaCaT cells can exhibit visible cellular changes on treatment. Green channel: plasma membrane & F-actin; red channel: mitochondria; blue channel: DNA.

each well holding a population of cells. The cells are given perturbants and then subsequently imaged with a microscope. In this study a perturbation means either a chemical treatment consisting of a small molecule at a specified concentration, or a CRISPR gene knockout. Cells can be fluorescently tagged for proteins or structures such as nucleus and mitochondria, and these form a sequence of image channels. The data output of an experiment is a large number of multi-channel images accompanied by metadata indicating the perturbation, plate and well of origin. Some molecules also have a known mechanism-of-action (MoA). MoA describes the pharmacological interaction which is believed to underlie the effect of a compound. MoA induced phenotypes are often subtle and require an algorithm to distinguish, but in some cases are visible by eye (Figure 1). Alongside perturbed samples, each plate includes controls expected to elicit no response, typically DMSO in compound screens and non-targeting CRISPR reagents for genetic screens.

A common way to quantitatively analyse cellular images is by extracting handcrafted morphological features using

tools such as CellProfiler (Carpenter et al., 2006), followed by aggregation and dimensionality reduction for downstream analysis. With the advent of highly accurate deep learning algorithms, there has been growing interest in replacing handcrafted features with learned image representations derived from data (Ando et al., 2017; Caicedo et al., 2018; Lafarge et al., 2019; Krentzel et al., 2023). These approaches aim to produce embeddings that uncover phenotypic variation that is difficult to capture with predefined measurements and support tasks such as treatment clustering, outlier detection, and MoA inference.

Recent work has focused on training large models, vision transformers, on increasing dataset sizes. These come in several flavours, typically adaptations of either masked autoencoders (Kraus et al., 2024; Kenyon-Dean et al., 2024) or DINO (Yao et al., 2024; Haslum et al., 2024; Doron et al., 2023; Kim et al., 2025). Also popular has been contrastive learning based around augmentations (Perakis et al., 2021; Bushiri Pwesombo et al., 2025; Huang et al., 2025) or multimodality (Sanchez-Fernandez et al., 2023; Fradkin et al., 2024; Lu et al., 2025; Yu et al., 2025). Contrastive learning naturally frames the learning problem in terms of proximity for semantically similar samples, which fits well the underlying phenotype alignment goal of HCS.

A classic approach is weakly supervised learning (WSL) (Caicedo et al., 2018; Moshkov et al., 2024), often considered the most performant method for small datasets (Kraus et al., 2024). WSL works by training a treatment classifier and then extracting embeddings from the encoder. Identically treated cells exhibit variation in shape and texture leading to substantial heterogeneity within a population (Figure 1). One goal of leveraging treatment labels is to limit how much the model responds to this variation and instead learn features more representative of inter-treatment differences. However since the objective of a classifier is to distinguish classes it can also separate treatments of the same phenotype, or learn other unwanted correlations (Sypetkowski et al., 2023), raising the question of whether there is a more effective way to introduce metadata.

This study considers simple injections of metadata labels into the conceptually appealing contrastive losses, and proposes a variant called Spherical Phenotype Clustering (SPC) with a modified update mechanism allowing more class overlap. We show in contrast to prevailing large scale trends, that the described metadata injections allow contrastive losses trained on limited data to achieve state-of-the-art performance in several highly regarded microscopy benchmarks, and that this extends to the case of uncurated screens. We further show that the method can compete with models having much higher parameter counts despite employing a vastly smaller number of training images.

## 2. Spherical Phenotype Clustering

As discussed in the introduction, a conventional way to incorporate metadata during training is with WSL, which is a treatment classifier where the encoder is then used to extract embeddings. We summarize the main components as follows:

- A dataset $\{\boldsymbol{x}_1, \boldsymbol{x}_2, ..., \boldsymbol{x}_N\}$ where each $\boldsymbol{x}_i$ is an image and $N$ is the total number of images in the dataset. For each image there is an associated label $y_i$, an integer value between $1$ and the total number of classes $C$.

- A parameterized encoder function $f$ (the model) that takes an image and outputs a $D$ dimensional embedding vector, $\boldsymbol{z}_i = f(\boldsymbol{x}_i)$. In this work we use either a ResNet18 (He et al., 2016) or, for fine-tuning, a vision transformer (Dosovitskiy et al., 2020).

- A final linear layer that maps embedding vectors to class predictions comprising a weight matrix $W \in \mathbb{R}^{D \times C}$ and bias term $\boldsymbol{b} \in \mathbb{R}^C$, so that the predicted class is given by the maximum element of the vector $W^T \boldsymbol{z}_i + \boldsymbol{b}$.

From now on, we let dataset indexed quantities such as $\boldsymbol{x}_i$ instead be indexed by minibatch. During training the function $f$ is optimized by repeatedly sampling a minibatch of $M$ images from the dataset and updating parameters based on the gradient of the cross-entropy loss function

$$l_i = -\log \frac{\exp(W_{y_i}^T \boldsymbol{z}_i + \boldsymbol{b}_{y_i})}{\sum_{j=1}^C \exp(W_j^T \boldsymbol{z}_i + \boldsymbol{b}_j)}, \qquad (1)$$

where $W_j$ indicates the $j$th column of $W$ and can be viewed as a prototype embedding for class $j$. In this and the following expressions we have omitted that the loss is averaged over batch samples, so that the actual loss is $L = (1/M) \sum_{i=1}^M l_i$.

Since treatments are always known when conducting a biological experiment, the usefulness of $f$ in HCS is not in the direct ability to perform treatment classification, but rather being able to cluster similar treatments to deduce relationships between them. Following training, images are converted to vectors using $f$ and then aggregated (by computing the mean) to create a well or treatment-level profile, which is a vector of the same dimension as $\boldsymbol{z}_i$. The profiles are then compared with distance based metrics to assess how effectively they cluster according to a biologically meaningful label such as MoA. This is the sense in which the approach is described as weakly supervised - the metadata provides a weak label which helps to learn the biological grouping a perturbation belongs to. To give a concrete example, if we had a dataset with multiple mTOR inhibitor

compounds, then each is a class with many associated images, but they belong to the same biological mechanism, mTOR inhibitor; so a prominent mTOR cluster raises confidence new molecules with unknown mechanism would be correctly grouped.

## 2.1. Contrastive options for incorporating experimental metadata

The contrastive loss in popular algorithm SimCLR is called NTXent (Chen et al., 2020a). We define first the following similarity function:

$$\text{sim}(\boldsymbol{u}, \boldsymbol{v}) = \exp\left(\frac{1}{\tau} \frac{\boldsymbol{u} \cdot \boldsymbol{v}}{|\boldsymbol{u}||\boldsymbol{v}|}\right),$$

which is the exponential of the cosine similarity multiplied by the reciprocal of temperature $\tau$. And we further define an augmented version of an image in the dataset as $\boldsymbol{x}_i^a$ and the corresponding embedding $\boldsymbol{z}_i^a = f(\boldsymbol{x}_i^a)$. The expression for NTXent can then be written as

$$l_i = -\log \frac{\text{sim}(\boldsymbol{z}_i, \boldsymbol{z}_i^a)}{\sum_{j=1}^{M} [\text{sim}(\boldsymbol{z}_i, \boldsymbol{z}_j^a) + \text{sim}(\boldsymbol{z}_i, \boldsymbol{z}_{j \neq i})]},$$

where the notation $\boldsymbol{z}_{j \neq i}$ indicates the $i$th term is not included in the summation. This loss function encourages the embedding for an image to be closer to its augmentation than to embeddings for other images sampled from the dataset.

There is substantial heterogeneity in the cells of HCS images that isn't relevant to discovering relationships between perturbations. For example changes in orientation of cells, technical noise across wells, shape and texture differences that are typical of variation in control samples. A fully unsupervised model would learn a lot of that variation. An autoencoder must remember it to reconstruct the entire image in detail and contrastive models will pick up on it as an effective way to make sure an image and its augmentation is similar. We here adapt contrastive learning so that in addition to the instance-instance comparisons, we have centroid-like class prototypes that constrain samples to be clustered nearby. The idea is to "forget" information related to variation inside of the class represented by those prototypes and produce an embedding space more representative of inter-treatment variation.

In conventional contrastive learning, images are typically compared to their augmentations because without prior knowledge there isn't a better way to define a positive sample pair (Wang & Isola, 2020). Considering the class vectors used in WSL as a kind of sample representing a treatment prototype, we use them to replace augmentations in the following altered NTXent:

$$l_i = -\log \frac{\text{sim}(\boldsymbol{z}_i, W_{y_i})}{\sum_{j=1}^{M} [\text{sim}(\boldsymbol{z}_i, W_{y_j}) + \text{sim}(\boldsymbol{z}_i, \boldsymbol{z}_{j \neq i})]}. \quad (2)$$

Compared with Eq. (1) this formulation differs in the use of the cosine similarity, the presence of instance-instance terms and a summation that now runs over the elements in a minibatch rather than over the classes.

A summation running over $M$ rather than $C$ elements is of interest in HCS where the number of classes in a dataset can be very large. For example, if they had decided to use weakly supervised learning with wells as targets, a recent work (Kraus et al., 2024) would be training on the equivalent of $C = 93 \cdot 10^6$ classes. More generally the number of classes grows linearly with the size of the dataset. If we assume a (fixed) $k$ images per well, the number of wells is $N/k$. Treatments similarly grow with the size of the dataset, but are usually performed with multiple replicates $r$, so that we would have $N/kr$ treatment classes. Example values of $k$ and $r$ would be 160 and 12 for the HaCaT JUMP dataset described below.

The other contrastive loss function we consider is SupContrast (Khosla et al., 2020)

$$l_i = \frac{-1}{\alpha} \log \sum_{p \in P(i)} \frac{\text{sim}(\boldsymbol{z}_i, W_{y_p}) + \text{sim}(\boldsymbol{z}_i, \boldsymbol{z}_{p \neq i})}{\sum_{j=1}^{M} [\text{sim}(\boldsymbol{z}_i, W_{y_j}) + \text{sim}(\boldsymbol{z}_i, \boldsymbol{z}_{j \neq i})]},$$

where $\alpha = 2|P(i)| - 1$, $P(i)$ is the set of batch indices labelled with class $y_i$ and we have again adapted the original expression by inserting $W$.

The normalized embeddings of the cosine similarity lie on a unit $(D-1)$-sphere. Optimizing over this space implicitly means learning angular relationships between embeddings. One argument given over the softmax in Eq. (1) is that it is undesirable to use a radially sensitive optimization objective during training when downstream test metrics based on cosine distance ignore the radial component of an embedding (Wang et al., 2017). This is the case in HCS, where cosine distance is used extensively in nearest-neighbour derived metrics and low dimensional plots generated with t-SNE or UMAP. In other work it is observed that when a class is tightly clustered on a hypersphere, the embeddings lie on a polar cap, which is linearly separable from the rest of the embeddings (Wang & Isola, 2020). The reciprocal of the temperature, $1/\tau$, has sometimes been viewed as a radial hyperparameter, controlling the amount of surface area available for clustering (Ranjan et al., 2017).

To distinguish the effect of spherical embeddings and temperature from other aspects of the contrastive losses, we

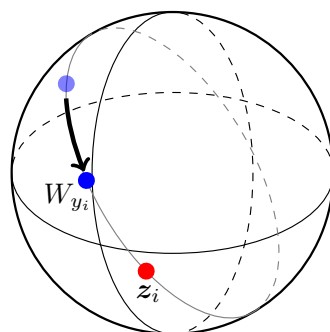

*Figure 2.* Geometric origin of class updates

also consider using an exact softmax with temperature and normalized vectors

$$l_i = -\log \frac{\text{sim}(\boldsymbol{z}_i, W_{y_i})}{\sum_{j=1}^{C} \text{sim}(\boldsymbol{z}_i, W_j)}, \qquad (3)$$

which is known as the $L_2$-constrained Softmax Loss (Ranjan et al., 2017; Wang et al., 2017)

## 2.2. Spherical update paths

It has been observed in previous work (Draganov et al., 2024; Wang et al., 2017; Zhang et al., 2020) that loss functions on spheres of the type considered above can exhibit convergence issues related to the unnormalized embeddings failing to lie on the unit sphere. In (Draganov et al., 2024) for example, it is shown that the gradient is orthogonal to a sphere, and then proven that the intuitive expectation finite step sizes along this gradient would lead to continually increasing embedding size holds and can slow convergence. It is also not clear that model parameters, which in the case of a ResNet are primarily filter kernel weights, and class vectors which play a role as embedding centroids, should be optimized under the same mechanism.

Motivated by this we introduce Spherical Phenotype Clustering (SPC) in which we will optimize class vectors by moving them along shortest spherical paths (geodesics) of the hypersphere. In order to define this trajectory we need a point for the class vector $W_{y_i}$ to move towards, and the obvious choice is $\boldsymbol{z}_i$, a model generated embedding for class $y_i$. Even with random initialization of $W$, optimization of the model quickly makes the $\{\boldsymbol{z}_i\}$ non-random. We now observe that a closer study of this mechanism yields the following result:

*The tangent to the shortest spherical path between $W_{y_i}$ and $\boldsymbol{z}_i$ is a negative multiple of the gradient of the cosine similarity loss function.*

A derivation of this relationship is given in Appendix A. Expanding the logarithm in NTXent Eq. (2) we have

$$l_i = -\log \frac{\text{sim}(\boldsymbol{z}_i, W_{y_i})}{\sum_{j=1}^{M} \left[ \text{sim}(\boldsymbol{z}_i, W_{y_j}) + \text{sim}(\boldsymbol{z}_i, \boldsymbol{z}_{j \neq i}) \right]}$$

$$= -\frac{1}{\tau} \frac{W_{y_i}^T \boldsymbol{z}_i}{|W_{y_i}^T||\boldsymbol{z}_i|} + \log \sum_{j=1}^{M} \left[ \text{sim}(\boldsymbol{z}_i, W_{y_j}) + \text{sim}(\boldsymbol{z}_i, \boldsymbol{z}_{j \neq i}) \right]$$

The first term is the cosine similarity loss multiplied by a constant factor, referred to as the attractive component of the loss because it decreases as $W_{y_i}$ and $\boldsymbol{z}_i$ draw close. Similarly, the second term has a repulsive effect. The observation that the gradient of the attractive component of the loss is proportional to the tangent of the geodesic allows us to unpack several differences between SPC and training with the NTXent of Section 2.1:

1. The arc path is confined to a specified radius, whereas gradient based NTXent updates move off the sphere.

2. A different learning rate and optimizer for class vectors. For example momentum may be desirable for model parameters but distort the spherical update path.

3. Repulsive terms are included in the model optimization objective, but not class vector updates.

Item 3 has a naive appeal in the clustering context. Many classes (treatments) would be expected to be very similar, and leaving out repulsive terms would allow them to pack together so that similar classes exhibit a greater degree of overlap. Conversely including the repulsive terms may lead to a model that separates treatments of the same MoA. This correspondence motivates us sometimes referring to SPC as NTXent w/o repulse when we wish to distinguish various ablations, though we always mean by this the modified NTXent of Eq. (2).

That the attractive component of NTXent is proportional to the cosine loss allows us to implement SPC by alternating two coupled updates on each minibatch. First, we update the encoder parameters using the full NTXent objective in Eq. (2): each sample embedding $\boldsymbol{z}_i = f(\boldsymbol{x}_i)$ is pulled toward its class vector $W_{y_i}$ (attraction) while being pushed away from other class vectors $W_{y_j}$ in the minibatch and from other instance embeddings $\boldsymbol{z}_{j \neq i}$ (repulsion). Second, we update the class vectors using only the attractive term, so that each class vector is moved toward embeddings of samples from its class without any repulsive forces. Concretely, we perform a no-momentum SGD step on the cosine objective $L(W) = -\sum_{i=1}^{M} W_{y_i}^T \boldsymbol{z}_i / |W_{y_i}^T||\boldsymbol{z}_i|$, followed by for each class $c$ the projection $W_c \leftarrow W_c / |W_c|$ to keep class vectors on the unit sphere. This "no-repulse" class update allows phenotypically similar classes to overlap in embedding space while the encoder is still trained with repulsive terms to prevent trivial collapse (Appendix A).

*Table 1.* BBBC021 ablation results, mean $\pm$ s.d. over three training runs. mAP score is reported with both well and treatment profiles. NTXent and SupContrast refer to the altered versions incorporating metadata described in Section 2.1. Note that removing repulsive terms from model updates (Cosine loss row) results in trivial collapse.

| Loss function | mAP (well) | mAP (treatment) | NSC | NSCB |
|---|---|---|---|---|
| Softmax (WSL) | $.790 \pm .012$ | $.905 \pm .009$ | $.967 \pm .005$ | $.920 \pm .006$ |
| $L_2$-Softmax | $.803 \pm .001$ | $.882 \pm .002$ | $.932 \pm .016$ | $.923 \pm .021$ |
| NTXent | $\mathbf{.892 \pm .011}$ | $.931 \pm .007$ | $\mathbf{.974 \pm .005}$ | $\mathbf{.956 \pm .010}$ |
| SupContrast | $.882 \pm .008$ | $\mathbf{.935 \pm .006}$ | $\mathbf{.974 \pm .005}$ | $.952 \pm .006$ |
| Cosine | $.284 \pm .010$ | $.402 \pm .012$ | $.207 \pm .056$ | $.188 \pm .063$ |
| *Confine to sphere & split optimizers* | | | | |
| NTXent | $.890 \pm .009$ | $.934 \pm .010$ | $.970 \pm .016$ | $.949 \pm .012$ |
| NTXent w/o repulse (SPC) | $\mathbf{.924 \pm .003}$ | $\mathbf{.959 \pm .006}$ | $\mathbf{.980 \pm .000}$ | $\mathbf{.967 \pm .010}$ |
| SupContrast | $.894 \pm .004$ | $.932 \pm .016$ | $.975 \pm .020$ | $.961 \pm .007$ |
| SupContrast w/o repulse | $.900 \pm .008$ | $.948 \pm .004$ | $.974 \pm .014$ | $.956 \pm .010$ |

## 3. Results

We report results on three datasets: classic benchmark dataset BBBC021, two new screens of HaCaT cells and RxRx3-core. After training, all embedding vectors for evaluation images originating from a given well are mean averaged to create a single well-level embedding profile. Similarly, embeddings for a perturbation are averaged to create a perturbation-level profile. For the ResNet models we also apply a domain adaptation technique to the batch norm layers. Refitting batch norm layer statistics during evaluation has been known for some time in computer vision to be an effective way to reduce data distribution shift (Li et al., 2018; Xiao & Snoek, 2024; Liang et al., 2025) - in our case we use it to mitigate the effects of plate variation. The technique has been applied previously in HCS (Lin & Lu, 2022; Sypetkowski et al., 2023; Kraus et al., 2024). Except for RxRx3-core, we do not perform postprocessing.

Two metrics we work with are $k$-nearest neighbours with $k = 1$ (1-NN) and mean average precision (mAP). In both cases we use cosine distance as the underlying distance metric. In this context, the precision and recall values for mAP are calculated at different thresholds of distance from a sample. See (Kalinin et al., 2024) for a detailed discussion of mAP in HCS. Unless otherwise stated, in the following sections both 1-NN and mAP use MoA as ground truth labels. All models are trained using perturbation (treatment or gene KO ID) as class, and results are mean averaged over three training runs.

### 3.1. BBBC021

BBBC021 (Caie et al., 2010; Ljosa et al., 2012) is comprised of MCF-7 cells treated with 113 small molecules at eight different concentrations. Among the full dataset there is a curated subset of 103 treatments annotated with MoA based on visibility of phenotypes and prior literature. Most prior

work has focused on this portion of the data, which we also use for our own benchmarking.

#### 3.1.1. ABLATIONS

We follow (Caicedo et al., 2018; Moshkov et al., 2024) in partitioning cells so that 60% form training data, and we use the other 40% for evaluation. The traditional way to report performance on BBBC021 has been under the NSC (Not-Same-Compound) and NSCB (Not-Same-Compound-or-Batch) scores on treatment-level profiles. These are both 1-NN based metrics. In the NSC score, distances are only calculated between samples that have different compound labels, so that high scores require alignment between different compounds with the same MoA. NSCB additionally requires that neighbours come from a different batch.

We begin by examining the effect of replacing the conventional WSL loss function with alternatives discussed in Section 2.1 - a regular softmax, the $L_2$ variant, and the two contrastive options. For the loss functions with a temperature hyperparameter we test three values, $\tau = 0.1, 0.25, 1$ and in all cases find $\tau = 0.25$ to give the best performance on MoA based metrics. We train a softmax based loss with and without the bias term but find limited difference, and report without the bias term in Table 1 for a consistent setup. Details of the other variants are given in Appendix C.

We see (Table 1) a substantial metric gain using either of the contrastive objectives. In particular the well aggregated mAP score is $\approx 0.1$ higher than in the non-contrastive case, and the treatment-level mAP finds an increase $> 0.03$. Normalizing vectors and including the hyperparameter $\tau$ does not, by itself, result in improved metrics.

On the second half of Table 1 we find simply confining to a sphere and using two optimizers does not improve performance but with the exclusion of repulsive terms it does, particularly for NTXent which is strictly speaking the

*Table 2.* BBBC021: comparison with previous work

| Study | Type | NSC | NSCB |
|---|---|---|---|
| (Ljosa et al., 2013) | CellProfiler / Factor Analysis | 94% | 77% |
| (Singh et al., 2014) | CellProfiler | 90% | 85% |
| (Lafarge et al., 2019) | Autoencoder | 92.9% | 82.2% |
| (Ando et al., 2017) | Transfer Learning | 96% | 95% |
| (Lin & Lu, 2022) | Transfer Learning / BEN | 96% | 88% |
| (Janssens et al., 2021) | DeepCluster | 97% | 85% |
| (Perakis et al., 2021) | SimCLR | 97% | 94% |
| (Caicedo et al., 2018) | WSL | 97% | 86% |
| (Cross-Zamirski et al., 2022) | WS-DINO (treatment) | 92% | 90% |
| Ours | SPC | **98.0%** | **96.7%** |

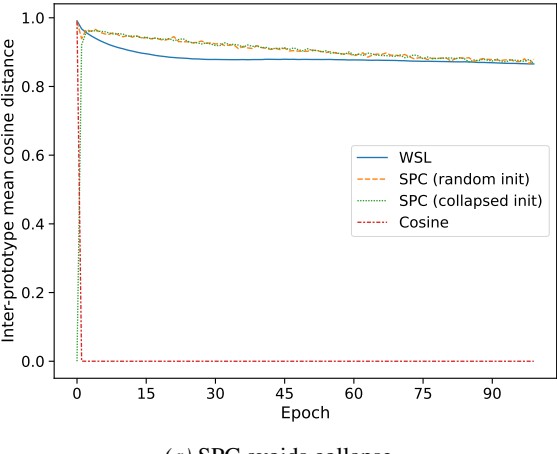

*(a)* SPC avoids collapse

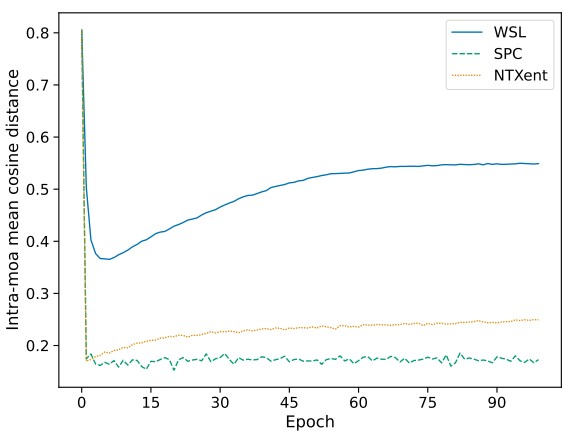

*(b)* Prototype alignment within MoA groups

*Figure 3.* Arguments made in the text around improved MoA clustering and avoiding collapse are empirically supported by cosine distances between prototypes during training.

case where our earlier derivation was valid. From here we continue with the NTXent w/o repulse setup (SPC).

Methods which use treatment metadata during training have described themselves as weakly supervised (Caicedo et al., 2018) or unsupervised (Janssens et al., 2021). In Table 2 we compare SPC to other results using metadata up to treatment level. SPC gives improved NSC and NSCB scores comparative to extant methods. Comparing with the closest analogue, the treatment classifier (Caicedo et al., 2018), we find +1% NSC and +10.2% NSCB.

### 3.1.2. PROTOTYPE COSINE DISTANCES

In motivating the contrastive loss and removal of repulsive terms, we suggested that this would enable treatment prototypes of the same MoA to cluster more closely than under the classification objective WSL. We also made the claim that our encoder-only repulsion objective would not be subject to mode collapse. Here we study the cosine distances between prototypes during training to establish empirically

whether these claims hold.

Figure 3a plots the mean cosine distance between prototypes during training. It is seen that SPC records cosine distances similar to that in WSL, and is clearly distinguished from the collapse situation illustrated by using cosine model loss. Even in the case where SPC is initialized with all vectors at a point, after one epoch the method has escaped collapse and trains successfully.

To examine how closely packed prototypes are for a given MoA, we also provide inter-prototype distances calculated only between prototypes of the same MoA (intra-moa), so that a lower average indicates better alignment. Figure 3b shows a wide differential between WSL which converges to ∼0.54 and SPC to ∼0.17 (and NTXent 0.25). We find earlier arguments are supported by prototype training dynamics. Looking at this graph, one may wonder if WSL or NTXent have peak performance at an early epoch. Figure 8 in Appendix C shows this not to be the case, and that performance increases over the duration of training.

*Table 3.* HaCaT cells: 1-NN/mAP scores on well profiles, mean ± s.d. over three training runs.

| Method | Screen | Unseen cells | | Unseen plates | | Unseen perturbations | |
|---|---|---|---|---|---|---|---|
| | | 1-NN | mAP | 1-NN | mAP | 1-NN | mAP |
| CellProfiler | JUMP | .327 | .115 | .271 | .113 | .330 | .196 |
| SimCLR | JUMP | .249 ± .004 | .078 ± .001 | .192 ± .007 | .080 ± .000 | .321 ± .046 | .141 ± .000 |
| WSL | JUMP | .482 ± .008 | .164 ± .003 | .398 ± .004 | .152 ± .000 | .461 ± .010 | .240 ± .010 |
| SPC | JUMP | **.496 ± .002** | **.207 ± .001** | **.420 ± .004** | **.194 ± .002** | **.525 ± .002** | **.298 ± .004** |
| CellProfiler | CRISPR | .550 | .197 | .522 | .246 | .755 | .243 |
| SimCLR | CRISPR | .403 ± .003 | .098 ± .000 | .388 ± .004 | .116 ± .001 | .598 ± .001 | .163 ± .001 |
| WSL | CRISPR | .626 ± .003 | .326 ± .002 | .545 ± .009 | .318 ± .003 | .772 ± .003 | .370 ± .009 |
| SPC | CRISPR | **.714 ± .014** | **.450 ± .004** | **.584 ± .017** | **.410 ± .013** | **.804 ± .008** | **.483 ± .011** |

## 3.2. HaCaT datasets

To explore a more challenging, real-world scenario we evaluate on two new cell painting screens, where the resultant images have not been curated. We adopt HaCaT cells, a spontaneously immortalised human keratinocyte cell line, and select two perturbant modalities: small molecules and genetic CRISPR knockouts. The JUMP-MoA small molecule library was initially developed by the Joint Undertaking for Morphological Profiling (JUMP) Cell Painting Consortium and aims at generating a large publicly available cell painting dataset (Chandrasekaran et al., 2023). This library comprises 90 compounds, 4 replicates per plate, covering 47 distinct MoA classes (Cimini et al., 2023), and thus serves as a suitable choice to elicit a broad range of cellular phenotypes. For CRISPR perturbations we selected 207 genes anticipated to produce distinct phenotypes across the key cell painting channels, including those linked to mitochondrial function, Golgi apparatus, F-actin, and nuclear division/cell cycle. Chemical screening was performed at three compound concentrations (0.1, 1.0 and 10 μM) with three replicate plates per concentration, while the genetic screen was performed with 5 replicate plates. Images were acquired in 4 channels at 40X magnification, 10 fields of view per well.

Training and evaluation are carried out with three different approaches to test splits. In the first case, cells, we use a 60:40 image split similarly to the BBBC021 split. Then we hold back plates, one of each concentration for JUMP, and two of the five for CRISPR. Finally we hold back perturbations, for JUMP keeping back all 3 plates of 1μM treatments, and for CRISPR holding back a random 40% of KOs as well as 40% of control wells. In this way we verify the model can learn similarity between perturbations not seen during training.

We compare with CellProfiler, SimCLR and WSL. CellProfiler v4.2.5 is applied to segment cellular objects and measure approximately 1500 morphological features. This

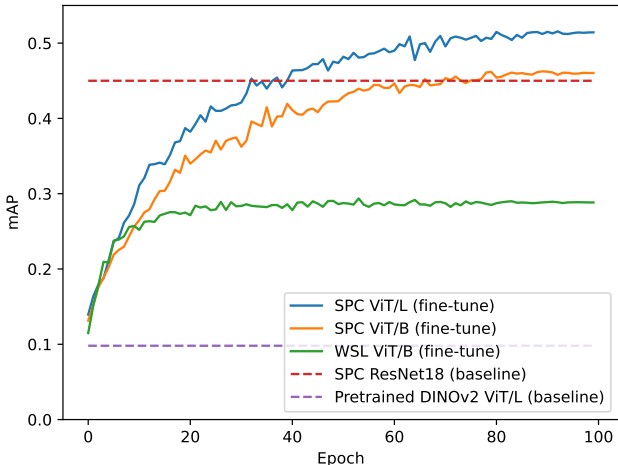

*Figure 4.* Performance of fine-tuned ViT models against baseline SPC and pretrained DINOv2.

was followed by aggregating to a well median, feature selection, and robust z-scoring to the plate DMSO control median and median-absolute-deviation values. WSL and SimCLR are trained with the same hyperparameter configuration as SPC. On these datasets we find it advantageous to train SPC with $\tau = 0.1$. We report results in Table 3. Numbers for the JUMP dataset are a better estimate of model performance because we randomized well positions to account for the confounding effects of plate layout and ran multiple treatments for each MoA. For CRISPR we have only the KO label, so results reduce to perturbation metrics and furthermore we used a typical CRISPR screen layout with each perturbation represented by a single well per plate rather than four randomized positions. Nonetheless, in all cases SPC results in a clear improvement.

### 3.2.1. FINE-TUNING DINOv2 MODELS

In the section on BBBC021, we compared with WS-DINO (Cross-Zamirski et al., 2022). WS-DINO uses WSL as

*Table 4.* RxRx3-core metrics for biological recall and drug-target interaction. Each biological recall column is a different database. Highest score **bolded**, and second highest underlined. *: Data from (Kraus et al., 2025). **: Data from (Huang et al., 2025). †: Mean over three training runs, individual runs reported in Appendix C.

| Method | Parameters | Biological recall | | | | Drug-target interaction | |
|---|---|---|---|---|---|---|---|
| | | CORUM | HuMAP | Reactome | StringDB | AP | Z-score↑ |
| Random embeddings* | - | .107 | .111 | .107 | .115 | $.214 \pm .003$ | 0.00 |
| CellProfiler* | - | .361 | .444 | .160 | .330 | $.276 \pm .018$ | 3.34 |
| OpenPhenom-S/16* | 25M | .300 | .352 | .158 | .281 | $.274 \pm .017$ | 3.89 |
| Phenom-1* | 307M | .395 | .482 | .188 | .349 | $.290 \pm .017$ | 4.35 |
| Phenom-2* | 1,860M | **.486** | **.553** | .197 | **.415** | $\mathbf{.307 \pm .015}$ | **6.04** |
| ViT-WSL** | 22M | .249 | .290 | .148 | .242 | $.259 \pm .013$ | 3.37 |
| MolPhenix** | 36M | .262 | .306 | .142 | .241 | - | - |
| CLOOME** | 25M | .328 | .406 | .135 | .278 | - | - |
| CellCLIP** | 1,477M | .354 | .416 | .145 | .307 | $.257 \pm .015$ | 2.81 |
| CWA-MSN** | 22M | .386 | .447 | .158 | .327 | $.267 \pm .015$ | 3.55 |
| SPC-ResNet18 (ours) † | 11M | .428 | .498 | **.204** | .382 | $.298 \pm .016$ | 4.31 |

a fine-tuning method for DINO trained vision transformers. We now ask the question of whether we outperform WSL as a fine-tuning mechanism, and also if there is any improvement over the trained-from-scratch ResNet18 models. To do so we work with the cell holdout CRISPR HaCaT dataset, and leverage pretrained DINOv2 (Oquab et al., 2023) models. LoRA (Hu et al., 2022) allows training of a low-dimensional subspace of the linear layers inside a transformer, helping to avoid overfit on smaller datasets. We train ViT/B DINOv2 models for WSL and SPC, applying LoRA with subspace dimension $r = 16$ to both. SPC outperforms WSL and both converge without overfit (Figure 4). Increasing to a ViT/L enables a small performance improvement over the fully trained ResNet18. An analogous experiment on the JUMP HaCaT data shows similar fine-tuning trends (Appendix C, Figure 14).

### 3.3. RxRx3-core

Among the vision transformer methods that have been investigated for HCS recently, one of the most celebrated has been OpenPhenom (Kraus et al., 2024). In (Kraus et al., 2024) it is shown by progressively training on larger and larger supersets of the RxRx3 HUVEC cell dataset, performance of a masked autoencoder (He et al., 2022) scales accordingly on the biological database relationship recall metric proposed in (Celik et al., 2022), and can outperform WSL. In a follow up article, the OpenPhenom group released to the community a subset of RxRx3 called RxRx3-core, intended as a benchmark dataset (Kraus et al., 2025). RxRx3-core comprises 222,601 images of 736 CRISPR KOs and 1,674 compounds over 8 concentrations. They evaluate on this dataset both the publicly available OpenPhenom model as well as the larger Phenom-1, and the ∼2B parameter Phenom-2 introduced

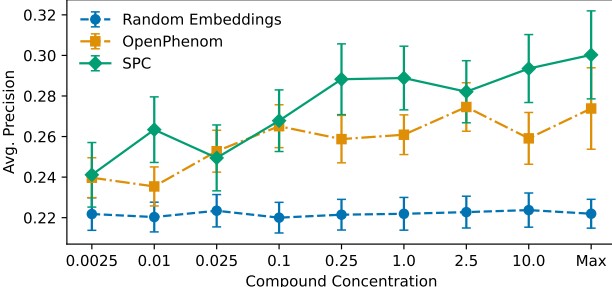

*Figure 5.* Mean average precision of OpenPhenom and SPC at different concentrations, with ± s.d. over 100 random seeds of negative samples.

in (Kenyon-Dean et al., 2024). OpenPhenom is trained on the full RxRx3 dataset, which is approximately 160x the size of RxRx3-core and contains additional perturbations as well as many more crops from the same cell populations (same source image). Phenom-1 and Phenom-2 are trained on larger datasets around 40x and 7x the size of RxRx3 respectively.

We train SPC on RxRx3-core, and evaluate two zero-shot prediction tasks using the evaluation code provided by (Kraus et al., 2025), including applying the same PCA-CS postprocessing step. Both tasks use perturbation profiles. The first is the biological recall metric by (Celik et al., 2022) mentioned above, where for each gene it is calculated the proportion of known relationships which are recalled by examining the top and bottom 5% of cosine distances to other perturbation profiles. The second metric is a drug-target interaction score using average precision to estimate how effectively the model ranks cosine similarity of a compound's known target genes against a set of genes selected

at random.

Results are presented in Table 4. We find higher performance (all metrics except Z-score) than Phenom-1 which is over 25x as large and trained on the full RxRx3, as well as CellProfiler, OpenPhenom and the recent contrastive methods in the second part of the table. Phenom-2 is a private model trained on a very large collection of HUVEC cells. Even with access to this model and compute, in many scenarios it would be unrealistic to be able to train on so many cells of one type, leaving a model which can perform effectively with limited data such as SPC as the best choice.

## 4. Related Work

Contrastive learning is widely used in unsupervised representation learning. Popular realizations include SimCLR (Chen et al., 2020a) and MoCo v2 (Chen et al., 2020b). Various methods implement image clustering in a contrastive setting. This can be explicit as with Contrastive Clustering (Li et al., 2021) or implicit in the sense that the technique involves refining models that are trained with contrastive learning (Van Gansbeke et al., 2020; Niu et al., 2022). Applying contrastive learning in the labelled data setting is a motivating theme of SupContrast (Khosla et al., 2020). A study of augmentations in contrastive learning with cell microscopy as an application was conducted in (Bendidi et al., 2023).

The $L_2$-softmax loss, and extensions involving margins and subclusters, have been studied in facial recognition (Ranjan et al., 2017; Wang et al., 2017; 2018). In these works it is typically shown that the explicit spherical geometry outperforms a conventional softmax in image classification tasks. One variant, ArcFace (Deng et al., 2019), was previously considered in HCS (Yang et al., 2022). HCS and facial recognition share a link in both being supplied with large numbers of classes with limited representation. Approximating a softmax with a contrastive loss in the large class setting has been found useful or necessary in the past (Mnih & Kavukcuoglu, 2013; Wu et al., 2018). The approach in (Wu et al., 2018) shows contrastive losses to be effective in the limit of each image having its own class vector representation, in this case the $> 10^6$ images of ImageNet.

Our approach is related to spherical interpolation which has a history in computer graphics (Shoemake, 1985). NTXent, our loss function of choice, was named in (Chen et al., 2020a) but appears in earlier work (Wu et al., 2018; Sohn, 2016) sometimes being referred to as NCELoss.

## 5. Conclusion

A notable feature distinguishing consumer images from those generated in HCS is some level of semantic labelling

always being present in the form of experimental metadata. Our work demonstrates this metadata can support contrastive objectives that outperform WSL and other popular methods on small datasets. By avoiding training with MoA and injecting only the chemical or genetic content of a well, we inherit the desirable property of WSL that the approach is viable in any arrayed screen regardless of how well annotated the content. The results on RxRx3-core indicate that modest computational resources can produce models that compete with or in some cases outperform current large scale microscopy machine learning, even when the data samples are drawn from many different experimental batches. This is of significant value to organisations participating in drug discovery or biological research.

## Impact Statement

This paper presents work whose goal is to advance the field of machine learning for scientific image analysis, specifically representation learning for high-content biological microscopy. The contributions are methodological and are intended to support research applications such as phenotypic screening and exploratory biological analysis. The work does not involve human-identifiable data, is not intended for clinical or decision-making use, and does not introduce ethical concerns beyond those already well understood for the application of machine learning in biological research. We therefore do not anticipate significant societal or ethical consequences specific to this work, beyond its contribution to established scientific workflows.

## Acknowledgements

We would like to thank Simon Boulton, David House, Katrin Rittinger, Jacob Bush, Paul Mercer and the rest of the Crick-GSK Prosperity Partnership for making this work possible. We also thank the Crick HPC team for their tireless efforts in providing a first class research computing platform at the Francis Crick Institute. This project was funded by a Prosperity Partnership grant from the Engineering and Physical Sciences Research Council (EPSRC), EP/V038028/1, by GSK and by the Francis Crick Institute which receives its core funding from Cancer Research UK, the UK Medical Research Council, and the Wellcome Trust.

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

# A. Theoretical arguments

## A.1. Correspondence between cosine similarity loss and spherical arc

In the notation that follows $\hat{z}$ denotes a unit vector in the direction of $z$. A convenient expression for the path along a spherical arc is provided by graphics algorithm SLERP (Shoemake, 1985). SLERP gives a spherically interpolated point between two vectors $w$, $z$ on a unit sphere, with $t \in [0, 1]$ indicating what proportion of distance along the arc the interpolated point should be

$$r(t) = \frac{\sin[(1-t)\Omega]}{\sin \Omega} \hat{w} + \frac{\sin[t\Omega]}{\sin \Omega} \hat{z}, \qquad \text{(A.1)}$$

where $\Omega$ is the angle between $\hat{w}$ and $\hat{z}$. It is a well known fact of high dimensional geometry that on an $n$-sphere, just as in the case of an ordinary sphere in three dimensional space, the shortest path between two points lies on a great circle. Great circles are defined by the intersection of the spherical surface with a 2-plane passing through the origin. On specifying a 2-plane using two points on the arc, the SLERP formula can be confirmed with straightforward geometry (Figure 6).

As described in the main article, the path we are interested in is the one starting at a class vector $w$ and ending at a model embedding vector $z = f(x)$ where image $x$ has the same class as $w$.

**Proposition.** *The tangent to the shortest spherical path between $w$ and $z$ is a negative multiple of the gradient of the cosine similarity loss function.*

The definition of the cosine similarity is $\cos \Omega = z^T w / |z||w|$ where $\Omega$ is the angle between $z$ and $w$. The cosine loss is the negative of this quantity. Computing the gradient of the cosine loss we get

$$\begin{aligned}
\nabla_w \left( -\frac{z^T w}{|z||w|} \right) &= -\frac{z}{|z||w|} + \frac{z^T w}{|z||w|^3} w \\
&= -\frac{z}{|z||w|} + \frac{|z||w| \cos \Omega}{|z||w|^3} w \\
&= -\frac{1}{|w|} (\hat{z} - \cos \Omega \, \hat{w}). \qquad \text{(A.2)}
\end{aligned}$$

Treating Eq. (A.1) as the path of a point over the sphere, we compute its tangent vector by taking the derivative with respect to $t$.

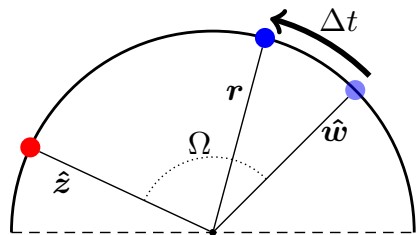

*Figure 6.* Verifying the SLERP formula

$$\begin{aligned}
\frac{dr}{dt}\bigg|_{t=0} &= \frac{d}{dt} \left\{ \frac{\sin[(1-t)\Omega]}{\sin \Omega} \hat{w} + \frac{\sin[t\Omega]}{\sin \Omega} \hat{z} \right\} \bigg|_{t=0} \\
&= \frac{\Omega}{\sin \Omega} \left( -\cos[(1-t)\Omega]\hat{w} + \cos[t\Omega]\hat{z} \right) \bigg|_{t=0} \\
&= \frac{\Omega}{\sin \Omega} \left( -\cos \Omega \, \hat{w} + \hat{z} \right),
\end{aligned}$$

which is proportional to the negative of the gradient in Eq. (A.2) when $\Omega / \sin \Omega \geq 0$. This is the case since we use the smaller arc on a great circle giving $0 \leq \Omega \leq \pi$. $\qquad \square$

## A.2. Encoder-only repulsion resists collapse

Consider loss function (2) in the main article with logarithm expanded

$$L = \sum_{i=1}^{M} \left( -\log \text{sim}(z_i, W_{y_i}) + \log \sum_{j=1}^{M} \left[ \text{sim}(z_i, W_{y_j}) + \text{sim}(z_i, z_{j \neq i}) \right] \right).$$

We provide a toy analysis with a batch size of two, so that we have two samples $z_1$ and $z_2$ with associated prototypes $W_{y_1}$ and $W_{y_2}$. Writing out terms explicitly gives

$$\begin{aligned}
L = &-\log \text{sim}(z_1, W_{y_1}) \\
&+ \log[\text{sim}(z_1, W_{y_1}) + \text{sim}(z_1, W_{y_2}) + \text{sim}(z_1, z_2)] \\
&- \log \text{sim}(z_2, W_{y_2}) \\
&+ \log[\text{sim}(z_2, W_{y_1}) + \text{sim}(z_2, W_{y_2}) + \text{sim}(z_2, z_1)].
\end{aligned}$$

Assume all prototypes exist at a point and the encoder is in an entirely collapsed state so that similarities are identical $s = \text{sim}(z_i, W_{y_j}) = \text{sim}(z_1, z_2)$. Then we have

$$\begin{aligned}
L_{collapse} &= -\log s + \log[s + s + s] - \log s + \log[s + s + s] \\
&= 2\log 3.
\end{aligned}$$

Now consider a mild perturbation of the encoder, so that $z_1$ and $z_2$ no longer sit exactly on top of the prototype vectors $W$. Assume for simplicity that they deviate the same distance from the collapsed $W$ so that we now have $\text{sim}(z_1, W_{y_1}) = \text{sim}(z_2, W_{y_2}) = \alpha s$ where $\alpha$ is some scalar value. We also have $\text{sim}(z_1, z_2) = \beta s$ where $\beta$ is another scalar. Then the new loss function becomes

$$L_{pert} = -\log \alpha s + \log[2\alpha s + \beta s] - \log \alpha s + \log[2\alpha s + \beta s]$$
$$= 2 \log[2 + \frac{\beta}{\alpha}].$$

Then as $\beta \to 0$ we have $L_{pert} \to 2 \log 2 < 2 \log 3 = L_{collapse}$ and more generally as long as $\beta < \alpha$ we have that $L_{pert} < L_{collapse}$. Remembering $s$ is a similarity, this implies that as long as the distance between $z_1$ and $z_2$ is greater than their distance to the prototypes, the perturbed loss function is smaller than at the collapsed encoder state. Since the perturbed samples can be made arbitrarily close to $W$ while maintaining this property, there must be local directions of the loss function smaller than the collapsed state, implying the existence of a gradient in descent directions. The prototype update will then move the $W_{y_i}$ toward the now separated $z$ inducing a separation in the previously collapsed $W$.

This analysis indicates that collapse is not a stable state for either the encoder or the prototypes, and that in such a state slight perturbations of the encoder lead to reduced loss and separated samples and prototypes. More generally because the prototype update moves prototypes towards the class-conditional batch mean, the prototypes would only collapse globally if the encoder collapsed first, but the preceding analysis suggests this is unlikely to occur.

With larger batch size, assuming $M < D$ and that all inter-sample similarities $\beta s$ are equal (for example sitting on the vertices of a regular simplex), then an almost identical analysis shows that $L_{pert} = M \log[M + (M - 1)\beta/\alpha]$ with $\beta = \alpha$ in the collapsed state, leading to the same conclusions as in the case with a batch size of two.

Note the importance of the instance-instance terms in providing a de-collapsing gradient. Without them, since the $W_{y_i}$ are all equal at collapse, any perturbation of the encoder would give

$$L = \sum_{i=1}^{M} \left( -\log \alpha_i s + \log \sum_{j=1}^{M} \alpha_i s \right) = M \log M.$$

So the encoder would not escape the collapsed state.

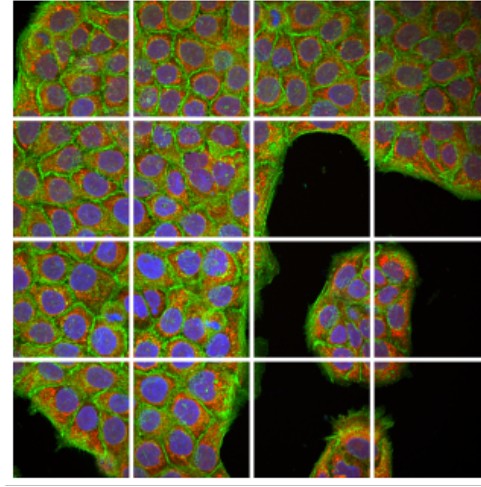

*Figure 7.* Raw 1080x1080 HaCaT images are resized to 512x512 and then tiled.

## B. Pre-processing and training details

We perform similar BBBC021 preprocessing to that in (Ando et al., 2017) including using the illumination correction functions (Singh et al., 2014), log transforming data and making use of the single cell crops defined in (Ljosa et al., 2013), but we rescale channels independently. To preprocess the HaCaT datasets we resized each image to 512x512 pixels and generated 16 crops under a 4x4 tiling scheme (see Figure 7). Since we have 10 fields per well the 4x4 tiling results in each well having 160 associated images of shape 128x128x4. We then filter the dataset by removing any tiles with very limited cellular content. We do this by calculating an intensity threshold in the nuclear channel over the whole dataset (10% of the maximum minus minimum intensities, post outlier clipping). Tiles where less than 5% of the pixels pass the threshold value are removed from the dataset. We now rescale each channel independently to lie in a 0 to 1 range after first clipping outlier intensities on a per channel basis. The outlier clipping thresholds we use are the 0.01 and 99.9 percentiles as in (Kim et al., 2025). Images are converted to uint8 for storage. RxRx3-core is largely already pre-processed - we tile the dataset 2x2 to generate four 256x256 crops as in (Kraus et al., 2025).

Our implementation of domain adaptation is similar to previous work (Li et al., 2018; Lin & Lu, 2022) but we do not make any changes to sampling during training. We use controls to compute the batch normalization statistics for BBBC021 and the HaCaT datasets, but for RxRx3-core where controls are not consistent across modalities we revert to the usual random sampling from batch.

All ResNet18 models used an embedding dimension 384. For BBBC021 we use the 3-channel augmentation scheme RandAugment with magnitude set to 12. For the multi-

channel HaCaT and RxRx3-core data we use three standard augmentations, RandomResizedCrop with scale (0.8, 1.0), RandomHorizontalFlip with probability 0.5, and Gaussian-Blur with kernel size 9, sigma 5, probability 0.5. We use the torchvision package to implement augmentations. Images in uint8 form are standardized by converting to float32, subtracting 127.5 from each pixel and then dividing by 255. BBBC021 and HaCaT were trained with the Adam optimizer (Kingma & Ba, 2014) set to initial learning rate of 0.0003 and no weight decay, a batch size of 16 and SPC SGD learning rate 0.1. RxRx3-core was trained with the same hyperparameter configuration as the HaCaT dataset, except we increase the batch size to 512 so that training completes in a reasonable amount of time. Vision transformers were trained with AdamW (Loshchilov & Hutter, 2017) initial learning rate 0.0003, weight decay 0.05 and betas (0.9, 0.95), SPC SGD initial learning rate 0.1, a cosine annealing scheduler with 5 warmup epochs, embedding dimension 768, a batch size of 256 and $\tau = 0.1$. Training runs were terminated after 100 epochs without early stopping.

The majority of training and evaluation was performed on a 32GB memory NVIDIA V100 GPU with a 20-core Intel Xeon Gold 6148 CPU and 256GB RAM. Vision transformer and RxRx3-core training runs were performed on an NVIDIA H100 GPU.

# C. Further results

## C.1. BBBC021

We collect in Table 5 extended results for BBBC021, including the temperature ablation study. Replicate metrics quantify the extent to which the closest samples in a well profile's neighbourhood are other wells of the same treatment. An interesting phenomenon emerges of replicate metrics falling on increase of MoA scores. Replicate based metrics penalize a sample being next to a different compound of the same MoA, while MoA metrics require the treatment to be close to other compounds of the same MoA. In support of the arguments we made in the main article, we find the exact softmax functions including the $L_2$ version generally better at separating a treatment from the rest of the dataset, but less effective at creating biological clusters of similar treatments. To some extent this is visible qualitatively (Figure 9). Table 5 also shows that ablating the domain adaptation and SGD learning rate components of SPC leads to the same conclusions reached in the main article, that a contrastive objective and dropping repulsive terms improves biological clustering.

Figure 8 provides the mAP score during training of WSL, SPC and NTXent, illustrating progression on the main metric without overfitting, and convergence of SPC to higher numbers over three training runs.

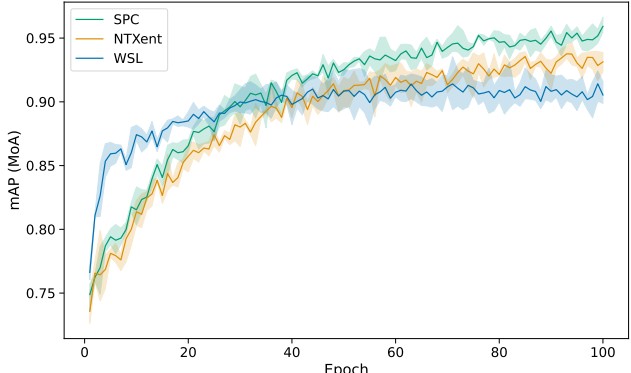

*Figure 8.* BBBC021 evaluation data, treatment profile MoA mAP during training with ± standard deviation across three runs.

## C.2. HaCaT datasets

### C.2.1. QUALITATIVE OBSERVATIONS

We now train a model on both the JUMP and CRISPR datasets, using the cell train splits of each dataset, and investigate the evaluation profiles for relationships between perturbations. A t-SNE (Van der Maaten & Hinton, 2008) of well profiles (Figure 10) shows consistent clustering of multiple replicates for a given treatment, and also local structure such as concentration gradients and multiple compounds of the same MoA in adjacent mini-clusters. Some cluster arrangements violate MoA. In one we find three compounds marked with different MoA: PD-198306 (MAP kinase inhibitor), neratinib (EGFR inhibitor), selumetinib (MEK inhibitor). An explanation is offered by all three lying on the MAPK/ERK signalling pathway (Orton et al., 2005). Inspection of the images gives marginal visual confirmation (Figure 11). The circumstance is clearer if we apply TGF-beta stimulation to the cells. Under these conditions the cells develop a characteristic staining pattern in the mitochondrial (red) channel. These observations suggest the algorithm is able to detect subtle cellular changes that visual inspection would agree with under more extreme conditions, and also illustrate the difficulty in effectively quantifying performance on datasets that are not curated in the manner of BBBC021.

Aurora kinase inhibitor compounds interfere with chromosome segregation during mitosis and have a very clear visual phenotype (Figure 1d). Examining the t-SNE we find clustering nearby CRISPR KOs of Aurora A kinase (AURKA) and ANLN. ANLN is also involved in mitosis, presenting a similar phenotype consisting of enlarged, multinucleated cells. To confirm the t-SNE is a faithful representation of model derived distances, we use perturbation profiles to compute the cosine distances between aurora kinase inibitor AMG900 and all perturbations in both datasets (Figure 12). The three unhighlighted perturbations closer than AURKA

*Table 5.* Extended BBBC021 results

| Method | Well profile | | | | Treatment profile | | |
|---|---|---|---|---|---|---|---|
| | MoA 1-NN | Replicate 1-NN | Replicate mAP | MoA mAP | MoA mAP | NSC | NSCB |
| *Loss function variants* | | | | | | | |
| (WSL) Softmax | 0.995 | 0.818 | 0.669 | 0.790 | 0.905 | 96.7% | 92.0% |
| Softmax w/ bias | 0.994 | 0.812 | 0.657 | 0.792 | 0.911 | 94.4% | 92.7% |
| $L_2$-Softmax ($\tau = 0.1$) | 0.992 | **0.931** | **0.847** | 0.666 | 0.789 | 82.2% | 78.6% |
| $L_2$-Softmax ($\tau = 0.25$) | 0.988 | 0.701 | 0.496 | 0.803 | 0.882 | 93.2% | 92.3% |
| $L_2$-Softmax ($\tau = 1.0$) | 0.797 | 0.566 | 0.10 | 0.292 | 0.388 | 40.1% | 30.7% |
| NTXent ($\tau = 0.1$) | 0.989 | 0.709 | 0.495 | 0.824 | 0.882 | 91.2% | 91.3% |
| NTXent ($\tau = 0.25$) | 0.995 | 0.692 | 0.457 | 0.892 | 0.931 | 97.4% | 95.6% |
| NTXent ($\tau = 1.0$) | 0.981 | 0.646 | 0.373 | 0.841 | 0.858 | 91.2% | 79.7% |
| SupContrast ($\tau = 0.1$) | 0.992 | 0.697 | 0.462 | 0.826 | 0.894 | 93.8% | 92.0% |
| SupContrast ($\tau = 0.25$) | 0.993 | 0.686 | 0.453 | 0.882 | 0.935 | 97.4% | 95.2% |
| SupContrast ($\tau = 1.0$) | 0.971 | 0.633 | 0.345 | 0.785 | 0.822 | 91.2% | 76.8% |
| Cosine | 0.680 | 0.517 | 0.096 | 0.284 | 0.402 | 20.7% | 18.8% |
| *Confine to sphere & split optimizers ($\tau = 0.25$, $lr = 0.1$)* | | | | | | | |
| NTXent | 0.996 | 0.693 | 0.460 | 0.890 | 0.934 | 97.0% | 94.9% |
| (SPC) NTXent w/o repulse | 0.996 | 0.669 | 0.425 | **0.924** | **0.959** | **98.0%** | **96.7%** |
| SupCon | 0.993 | 0.700 | 0.476 | 0.894 | 0.932 | 97.5% | 96.1% |
| SupCon w/o repulse | 0.992 | 0.662 | 0.409 | 0.900 | 0.948 | 97.4% | 95.6% |
| *SGD learning rate ablations* | | | | | | | |
| SPC ($lr = 0.01$) | 0.995 | 0.668 | 0.421 | 0.907 | 0.947 | 97.4% | 96.7% |
| SPC ($lr = 1$) | 0.994 | 0.646 | 0.393 | 0.909 | 0.939 | 97.7% | 96.3% |
| *No batch norm domain adaptation* | | | | | | | |
| WSL | 0.998 | 0.867 | 0.753 | 0.708 | 0.880 | 87.7% | 92.0% |
| SPC | **1.000** | 0.720 | 0.537 | 0.903 | 0.918 | 88.0% | 96.0% |

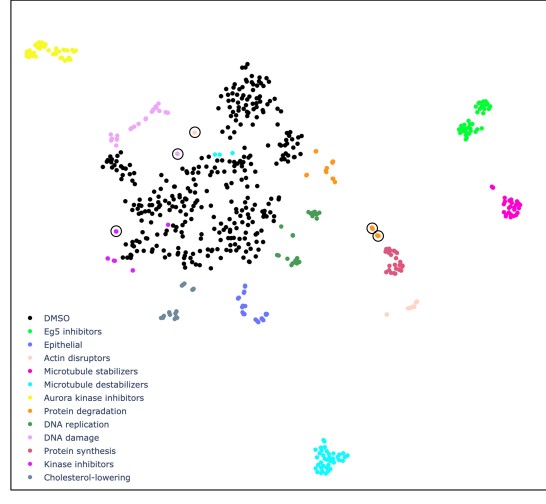

*(a) $L_2$-Softmax ($\tau = 0.1$)*

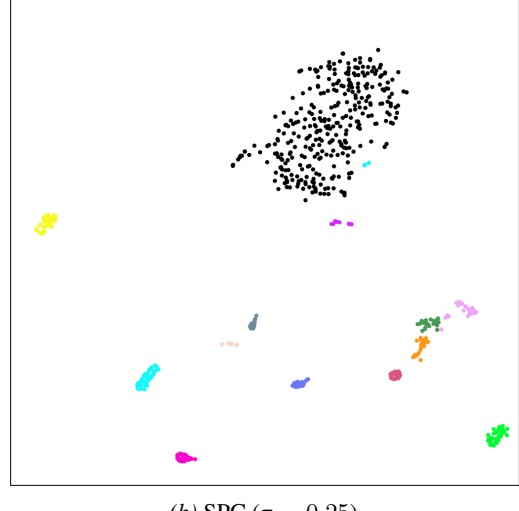

*(b) SPC ($\tau = 0.25$)*

*Figure 9.* High performance on replicate matching doesn't necessarily improve biologically meaningful clustering. Circled in the $L_2$-Softmax plot are instances where all treatment replicates are tightly packed, but detached from a main MoA cluster. By contrast SPC MoA clusters are well defined, even if there is internal overlap of treatments. Plots generated with TSNE on evaluation well profiles.

are CRISPR KOs of CCNA2, CDK1 and GOLGA2 which are closely related genes also involved in cell cycle and mitosis. For example GOLGA2 activates AURKA through its interaction with TPX2 (Wei et al., 2015). It is seen that we are able to effectively deduce which gene KOs have

similar effects to the AMG900 molecule. This type of tight clustering between chemical and genetic perturbations is an open challenge, and more generally even a small improvement in the ability to match chemicals to known phenotypic landmarks can greatly impact drug discovery by enabling

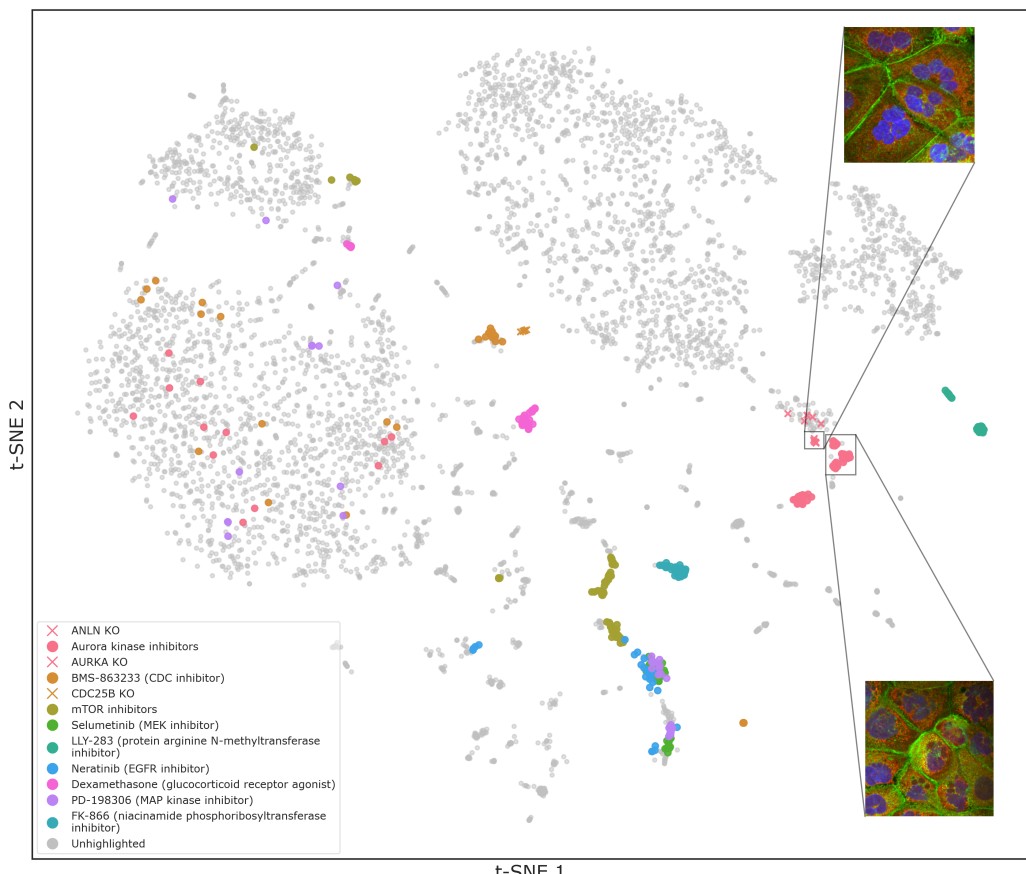

*Figure 10.* SPC aligns Aurora kinase inhibitor molecules with associated gene KOs, among other phenotypic clusters. t-SNE of well profiles, with several arrangements highlighted. Highlighted unclustered wells are typically low concentration treatments. Insets: ANLN KO and AMG900.

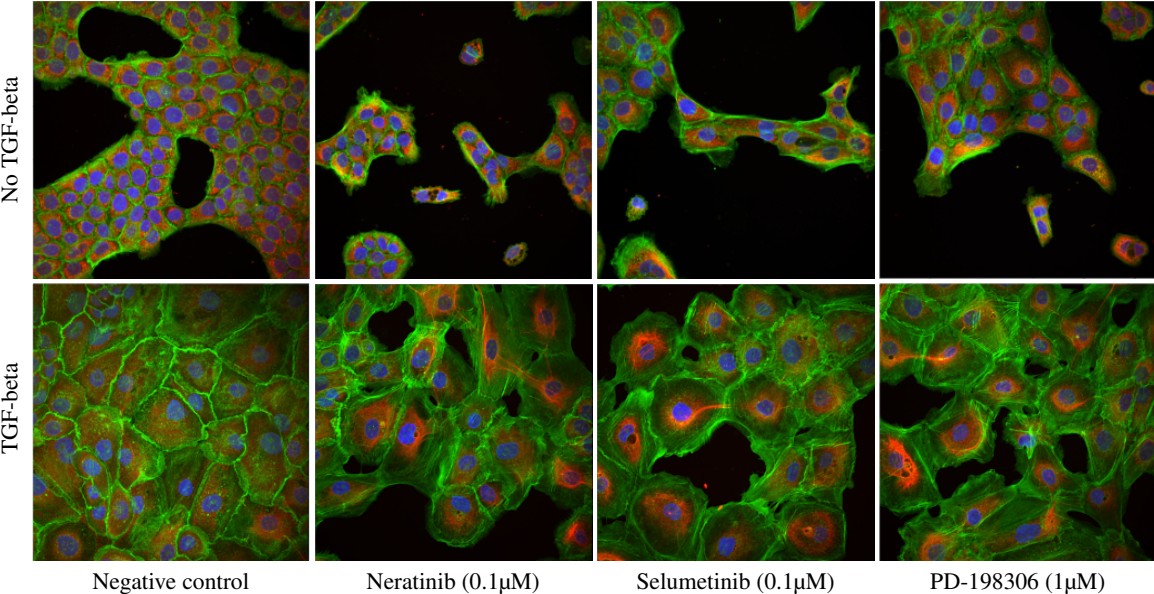

*Figure 11.* Visual similarity between three compounds with different MoA that SPC identified as similar, and negative controls for comparison. Cells are also shown with TGF-beta stimulation where the mitochondrial response becomes prominent.

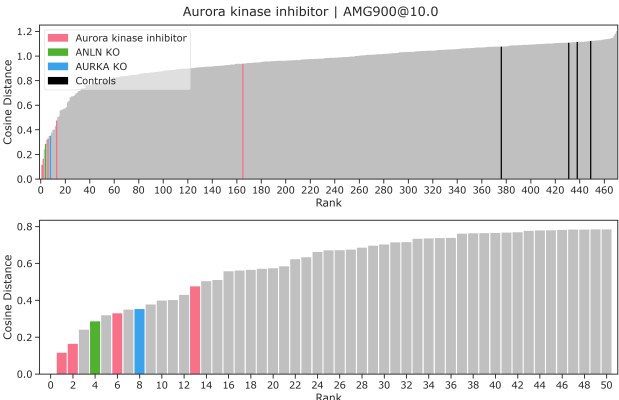

Figure 12. Ordered cosine distances between perturbation profiles and AMG900. First row: all profiles, second row: closest 50.

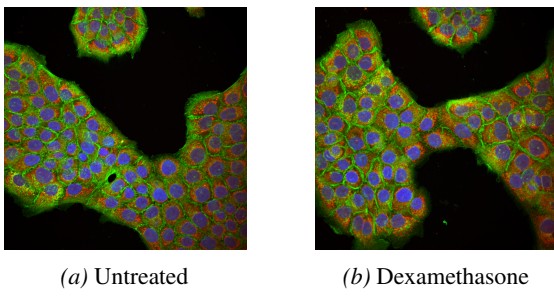

*(a)* Untreated      *(b)* Dexamethasone

Figure 13. Glucocorticoid receptor agonist dexamethasone is difficult to distinguish from control samples by inspecting images.

the assignment of unknown chemical entities to a specific mechanism of action, target, or biological function (Chandrasekaran et al., 2024).

We also find convincing qualitative clustering of the glucocorticoid receptor agonist dexamethasone. Dexamethasone is an example of a compound for which it is challenging to see any visually discernible effect (Figure 13), and furthermore we have found limited success extracting it with Cell-Profiler. Quantitatively the mAP for this MoA is 0.14 using CellProfiler and 0.64 for the model of Figure 10 illustrating that our deep learning method can identify phenotypes that might otherwise be missed.

### C.2.2. OUT-OF-DISTRIBUTION GENERALIZATION AND BATCH EFFECTS

Considering the extent to which SPC can make inferences about concepts that were not included in the optimization objective as metadata, we first note treatment information does not directly encode MoA or, for RxRx3-core, the relationships among genes and their interactions with molecules. Strong results on these tasks demonstrates that we've effectively learnt meaningful biological relationships that weren't injected during training.

Table 6. Out-of-distribution generalization on HaCaT screens.

| Method | Training data | Eval data | 1-NN | mAP |
|--------|---------------|-----------|------|-----|
| WSL | CRISPR | JUMP | .335 | .090 |
| SPC | CRISPR | JUMP | .429 | .119 |
| WSL | JUMP | CRISPR | .510 | .179 |
| SPC | JUMP | CRISPR | .550 | .227 |
| WSL | JUMP rerun | JUMP | .473 | .145 |
| SPC | JUMP rerun | JUMP | .511 | .178 |

Table 7. SPC batch effect quantification with HaCaT cells.

| Dataset | Graph conn. | Silhouette | KBET |
|---------|-------------|------------|------|
| JUMP 1 screen | .450 | .817 | .652 |
| JUMP 2 screens | .450 | .806 | .600 |

Nonetheless, it is also of interest to see how SPC generalizes to classes and batches not present during training. In the main article we reported for the HaCaT screens results with holdout plates and perturbations. We now take this a step further and report results training on one screen, JUMP or CRISPR, and evaluating on the other. This allows us to assess performance on a perturbation modality and experimental batch not seen during training (the CRISPR and JUMP screens were performed five months apart). Table 6 results show that SPC, unlike WSL, continues to outperform CellProfiler and SimCLR on an unseen perturbation modality (compare cell holdout column in Table 3).

These results, along with main article results on BBBC021 and RxRx3-core (which contains 180 experimental batches and 1744 plates) indicate a robustness to batch effects. We add a quantification of batch effects with a second HaCaT JUMP screen which has been prepared and imaged under the same conditions as the one described in the main article, except four months later, allowing us to isolate batch effects from other differences in screen content. Employing three metrics measuring batch effects (see (Arevalo et al., 2024)), graph connectivity, silhouette batch and KBET, we take plate as the batch variable and report scores in Table 7. It is seen that there is relatively little increase in batch effect metrics by incorporating a second experimental batch.

We further report results training on the JUMP rerun dataset and evaluating on the JUMP dataset included in the main article. On Table 6 we find performance in this regime is competitive to that on Table 3 suggesting there will be comparatively little performance loss when evaluating on a held out screen if similar phenotypes were seen during training.

*Table 8.* Further RxRx3-core results on biological database recall and drug-target interaction. All results in this table use a ResNet18 model. Except where indicated, perturbation was used as the training label.

| Method | Biological recall | | | | | Drug-target interaction | |
| --- | --- | --- | --- | --- | --- | --- | --- |
| | CORUM | HuMAP | Reactome | SIGNOR | StringDB | AP | Z-score↑ |
| $L_2$-Softmax($\tau = 0.01$) | .418 | .468 | .184 | .189 | .365 | $.295 \pm .017$ | 3.93 |
| $L_2$-Softmax($\tau = 0.1$) | .408 | **.501** | .189 | .189 | .364 | $.268 \pm .017$ | 2.48 |
| WSL | .414 | .494 | .195 | .189 | .371 | $.287 \pm .019$ | 3.16 |
| SimCLR | .371 | .410 | .181 | **.204** | .327 | $.251 \pm .015$ | 1.73 |
| SPC (well label) | .361 | .433 | .152 | .143 | .299 | $.263 \pm .018$ | 2.08 |
| SPC mean | **.428** | .498 | **.204** | .198 | **.382** | **$.298 \pm .016$** | **4.31** |
| SPC training run 1 | .423 | .511 | .221 | .200 | .386 | $.298 \pm .017$ | 4.09 |
| SPC training run 2 | .443 | .497 | .200 | .195 | .385 | $.298 \pm .015$ | 4.45 |
| SPC training run 3 | .420 | .487 | .193 | .200 | .376 | $.297 \pm .017$ | 4.40 |

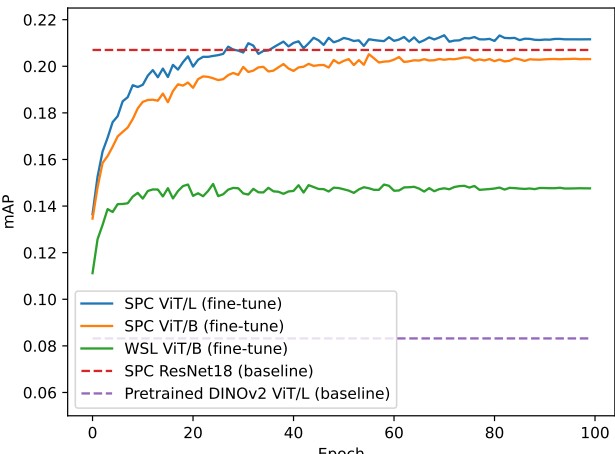

*Figure 14.* HaCaT JUMP data shows similar fine-tuning trends to CRISPR data (cell holdout).

### C.3. RxRx3-core

Table 8 reports results for individual SPC training runs. We also repeat the comparison with the exact softmax approaches tested on BBBC021 including the $L_2$ variety. And to help isolate the impact of the metadata on results we train a baseline SimCLR model and an SPC model using wells as targets. Examining Table 8 we conclude that neither the contrastive structure alone, nor the form of SPC without perturbation information is sufficient to produce the improved RxRx3-core scores. However, comparing with WSL and $L_2$ we find the perturbation metadata alone does not produce a model which clearly outperforms Phenom-1. This implies that both the perturbation metadata information and the SPC loss structure are important components in the improved performance on biological recall and drug-target interaction prediction.

More details on the metrics used in Table 4 and Table 8, including the definition of the Z-score, are given in (Kraus et al., 2025).

## D. HaCaT screening

We followed the standard cell painting assay protocol (Bray et al., 2016; Cimini et al., 2023), except for optimisations of dye concentrations to achieve optimal staining in Ha-CaT cells. For our staining procedures, MitoTracker was added live to cells, followed by cell fixation, permeabilisation and staining for the following subcellular compartments: nuclear DNA (Hoechst 33342), endoplasmic reticulum (concanavalin Alexa-488), RNA and nucleoli (syto-14), Golgi apparatus and plasma membrane (wheat germ agglutinin Alexa-555), and F-actin (phalloidin Alexa-568). Eight chemical substitutions to JUMP-MoA were made due to intellectual property constraints and two due to supply issues. We treated JUMP for 3 days. CRISPR screening was performed using 10 nM guide RNA, treating for 5 days to ensure maximum gene knockdown efficiency. Images were acquired using an Opera Phenix Plus high content microscope, and were max projected per channel.

# E. Algorithm pseudocode

---
**Algorithm 1** SPC training

---
**Require:** Dataset $\{(x_i, y_i)\}_{i=1}^N$, encoder $f_\theta$, class vectors $W \in \mathbb{R}^{D \times C}$, temperature $\tau$

**Require:** Encoder optimizer $\mathrm{Opt}_\theta$, prototype learning rate $\eta_W$

1: **while** not converged **do**
2:     Initialize encoder parameters $\theta$ and class vectors $W$
3:     Normalize each class vector: $W_c \leftarrow W_c / \|W_c\|_2$
4:     **for** each minibatch $\{(x_i, y_i)\}_{i=1}^M$ **do**
5:         Compute embeddings $z_i = f_\theta(x_i)$
6:         Normalize embeddings and prototypes: $\hat{z}_i \leftarrow z_i / \|z_i\|_2$, $\hat{W}_c \leftarrow W_c / \|W_c\|_2$
7:         Compute encoder loss $L$ (Eq. (2))
8:         Update encoder with full contrastive objective: $\theta \leftarrow \mathrm{Opt}_\theta(\theta, \nabla_\theta L)$
9:         Detach embeddings $\hat{z}_i$ from the computation graph

10:         Compute prototype-only attractive loss

$$\mathcal{L}_W = -\sum_{i=1}^M \hat{z}_i^\top \hat{W}_{y_i}.$$

11:         Update only prototypes appearing in the minibatch: $W_{y_i} \leftarrow W_{y_i} - \eta_W \nabla_{W_{y_i}} \mathcal{L}_W$
12:         Renormalize updated prototypes: $W_{y_i} \leftarrow W_{y_i} / \|W_{y_i}\|_2$
13:     **end for**
14: **end while**
15:
16: **return** Trained encoder $f_\theta$

---

