# OpenReview forum: "Asymmetric Contrastive Objectives for Efficient Phenotypic Screening"
_ICML.cc/2026/Conference — ICML 2026 regular_

### Official Review · Reviewer_NRiN · 2026-03-11

**Soundness:** 4
**Presentation:** 4
**Significance:** 4
**Originality:** 4
**Overall Recommendation:** 4
**Confidence:** 4

**Summary:**

This study seeks to assess the concept of incorporating experimental metadata as learned class prototype vectors within modified contrastive objectives for high-content microscopy screening. The authors intend to consider the concept of asymmetric hyperspherical updates (SPC) where class prototypes are attracted toward sample embeddings without repulsive forces, allowing phenotypically similar treatments to naturally overlap in embedding space while the encoder retains full contrastive training.

**Compliance With Llm Reviewing Policy:**

Affirmed.

**Final Justification:**

The authors have addressed the weaknesses I outlined adequately. I maintain my initial score of 4.

**Key Questions For Authors:**

Is there evidence that performance gains on biological recall are driven by specific metadata attributes (e.g., target identity versus compound class)?

**Limitations:**

Yes.

**Strengths And Weaknesses:**

S1: The paper is methodologically clean and well-motivated throughout.

S2: The geometric derivation connecting the attractive gradient to the geodesic on the hypersphere gives SPC a principled foundation rather than presenting it as a heuristic, and the intuition that repulsive class updates actively harm MoA clustering is compelling and well-supported by the ablation in Table 1.

S3: The progression from WSL to contrastive metadata injection to the asymmetric spherical update is logical and easy to follow.

S4: Evaluation rigor is a genuine strength (multiple datasets, perturbation modalities, and holdout strategies paint a convincing picture of real-world utility)

S5: The qualitative biological analysis (MAPK clusterin) meaningfully connects embedding geometry to biological discovery.


W1: The primary concern is that training and evaluation are performed on overlapping data of the same screen, a setting not widely stress-tested in the HCS representation learning literature. It would strengthen confidence in the method to see results in a fully held-out screen setting.

W2: Additionally, in the RxRx3-core evaluation there is no ablation isolating the contribution of metadata incorporation itself: a no-metadata baseline would clarify how much of the biological recall gain is attributable to the contrastive structure versus the prototype label signal specifically.

W3: Relatedly to the above, it would be valuable to understand which aspects of the treatment metadata most drive recall performance, since treatment labels encode heterogeneous information (chemical identity, concentration, target) and their relative contributions remain unexamined.

---

> ### Author Rebuttal · Authors · 2026-03-30
>
> We thank the reviewer for their feedback and helpful thoughts on clarifying the sources of performance improvement.
>
> ## Evaluating on unseen screens (W1)
>
> We discuss this topic in appendix C2.2, where we also evaluate on unseen HaCaT screens by training on JUMP and evaluating on CRISPR and vice-versa (JUMP and CRISPR were separate screens). Results are on table 6. In this held out screen setting SPC continues to outperform WSL, SimCLR and CellProfiler. We strengthen the result by reporting training a separate JUMP screen with the same perturbations (mentioned in appendix C2.2) and evaluating on the screen in the main article. The results below show performance competitive with that reported on table 3 suggesting there will be comparatively little performance loss when evaluating on a held out screen if similar phenotypes were seen during training.
>
> Method | 1-NN | mAP
>
> WSL | .473 | .145
>
> SPC | .511 | .178
>
> ## Metadata contribution to performance (W2, W3, key question)
>
> The treatment labels we used were a compound at a specific concentration, so that we do not explicitly tell the model that compounds at different concentrations are related. In order to help isolate the contribution of the metadata (and perturbation label specifically) to model performance, we provide two additional baselines for RxRx3-core. We add SimCLR with its closely related loss that does not use metadata, and we run SPC using well identity as the target label.
>
> Method | CORUM | HuMAP | Reactome | SIGNOR | StringDB | AP | Z-Score
>
> SimCLR | .371 | .410 | .181 | .204 | .327 | .251 ± .015 | 1.73
>
> SPC (wells) | .361 | .433 | .152 | .143 | .299 | .263 ± .018 | 2.08
>
> Examining these results we conclude that neither the contrastive structure alone, nor the form of SPC without perturbation information is sufficient to produce the improved RxRx3-core scores. However, comparing with WSL and others on table 8 in the appendix we also find the perturbation metadata alone does not produce a model which clearly outperforms Phenom-1. This implies that both the perturbation metadata information and the SPC loss structure are important components in the improved performance on biological recall and drug-target interaction prediction.
>
> For RxRx3-core there are many more chemical perturbations than genetic (13,358 vs 736), so it is not clear we would be able to disentangle contribution of perturbation modality from the skew in dataset composition. But the HaCaT unseen screen results give some impression of cross perturbation modality contribution to performance.
>
> We will add the extra results above to tables 6 and 8 in the appendix.

---

> > ### Author Rebuttal · Reviewer_NRiN · 2026-04-03
> >
> > The authors have adequately addressed my concerns with the results provided. My largest concern was W1, which the authors have pointed me to the appendix.

---

### Official Review · Reviewer_qP7b · 2026-03-12

**Soundness:** 3
**Presentation:** 3
**Significance:** 2
**Originality:** 2
**Overall Recommendation:** 4
**Confidence:** 4

**Summary:**

This paper proposes Spherical Phenotype Clustering (SPC), a method that injects experimental metadata
  (treatment labels) into contrastive loss functions for learning image representations from high-content
  screening (HCS) microscopy data. The key idea is twofold: (1) replace augmentation-based positive pairs in
   contrastive losses (NTXent, SupContrast) with learnable class prototype vectors derived from treatment
  metadata, and (2) decouple the optimization of class vectors from the encoder by updating class vectors
  using only the attractive component of the loss along geodesics of the unit hypersphere, while the encoder
   is trained with the full contrastive objective including repulsive terms. This asymmetric update
  mechanism allows phenotypically similar classes to overlap in embedding space rather than being pushed
  apart. The method is evaluated on three datasets, BBBC021, two new HaCaT cell screens (JUMP and CRISPR),
  and RxRx3-core, demonstrating improvements over weakly supervised learning (WSL), SimCLR, CellProfiler,
  and competitive performance against models 10–100x larger in parameter count (e.g., Phenom-1/2) despite
  using only a ResNet18 with 11M parameters.

**Compliance With Llm Reviewing Policy:**

Affirmed.

**Key Questions For Authors:**

1. What prevents class vector collapse under the no-repulse update? With only attractive updates, what
  mechanism ensures that class vectors for dissimilar treatments don't converge to similar regions of the
  sphere? Is it purely the encoder's repulsive training that maintains separation, or does the
  initialization/learning rate play a critical role? A clearer analysis (even empirical, e.g., tracking
  inter-class angular distances during training) would be very informative. A positive answer showing
  robustness would increase my confidence in the method's reliability.
  2. How does SPC perform with WSL-ResNet18 baseline on RxRx3-core? Table 8 in the appendix shows WSL
  results, but Table 4 (the main comparison table) does not include a WSL-ResNet18 row. Could you clarify
  the WSL-ResNet18 numbers for all RxRx3-core metrics? This is important for isolating the contribution of
  the SPC loss from other factors such as architecture, postprocessing.
  3. How sensitive is the method to batch size? The contrastive loss sums over minibatch elements rather
  than classes. Batch size is 16 for BBBC021/HaCaT but 512 for RxRx3-core. Does the relative performance of
  SPC vs. WSL change significantly with batch size? This matters for understanding whether the improvement
  comes from the loss formulation or implicitly from the negative sampling distribution.
  4. Can you provide wall-clock training time comparisons? The paper emphasizes compute efficiency, but only
   reports parameter counts. How does SPC's training time compare to WSL and SimCLR on the same hardware,
  especially given the two-step optimization (encoder + class vector projection per batch)?

**Limitations:**

yes

**Strengths And Weaknesses:**

Strengths

  1. Well-motivated geometric insight. The connection between the attractive
  component of NTXent and the geodesic tangent on the hypersphere (Appendix A) is elegant and provides a
  principled justification for the asymmetric update rule. Decoupling class vector updates from repulsive
  terms to allow phenotypically similar treatments to overlap is a simple but insightful design choice
  well-suited to the HCS domain where ground-truth groupings (MoA) are coarser than training labels
  (treatments).
  2. Strong empirical results with limited compute. Achieving competitive or superior
  performance to Phenom-1 (307M params, trained on 160x more data) and outperforming CellProfiler,
  OpenPhenom, and recent contrastive methods on RxRx3-core using only a ResNet18 trained on the core subset
  is a compelling demonstration of data and compute efficiency. The consistent improvements across three
  diverse datasets (BBBC021, HaCaT, RxRx3-core) strengthen confidence in the method's generality.
  3. Comprehensive evaluation protocol. The paper evaluates on a wide array of metrics (mAP,
  1-NN, NSC, NSCB, biological recall across four databases, drug-target interaction AP/Z-score) and multiple
   data split strategies (unseen cells, unseen plates, unseen perturbations). The inclusion of uncurated
  HaCaT screens alongside the classic curated BBBC021 benchmark adds practical relevance.
  4. Thorough ablation study. Table 1 systematically disentangles the
  contributions of (a) contrastive vs. softmax losses, (b) spherical confinement + split optimizers, and (c)
   removal of repulsive terms. This makes it clear that the performance gain is not simply from using a
  contrastive loss or normalizing embeddings, but specifically from the asymmetric class update mechanism.
  5. Practical applicability. The method requires only treatment metadata (always available
  in HCS), uses a small ResNet18, and is shown to work as a fine-tuning technique with LoRA on DINOv2 ViTs.
  This makes it directly usable in resource-constrained drug discovery labs without access to large-scale
  pre-training infrastructure.

  Weaknesses

  1. Limited theoretical analysis of the "no-repulse" mechanism. While the geodesic-gradient
  correspondence (Appendix A) is neat, it only justifies the direction of the attractive update. The paper
  does not analyze why removing repulsive terms from class vector updates specifically helps, e.g., under
  what conditions does this lead to better MoA clustering vs. trivial convergence of all class vectors to a
  single point? The cosine-only baseline (Table 1, "Cosine" row) shows that removing repulsive terms from
  both encoder and class vectors leads to collapse, but there is no formal analysis of when the asymmetric
  setup avoids this. A toy analysis or convergence guarantee would strengthen the contribution.
  2. Incomplete baselines on RxRx3-core. The comparison with Phenom-2 is somewhat unfair in both
   directions: Phenom-2 is trained on vastly more data (making SPC's competitive result impressive), but it
  is also a private model evaluated by others, so the comparison is not fully controlled. More importantly,
  the paper does not compare against WSL with the same ResNet18 backbone on RxRx3-core (Table 4 only shows
  ViT-WSL from another paper), making it hard to isolate the contribution of SPC vs. the backbone/training
  choices on this dataset. Table 8 includes WSL but only in the appendix.
  3. Sensitivity to hyperparameters not fully explored. The optimal temperature differs between
  datasets (τ=0.25 for BBBC021, τ=0.1 for HaCaT and ViT fine-tuning). The SGD learning rate for class
  vectors is ablated only on BBBC021 (Table 5). Given that HCS datasets vary widely in scale, number of
  classes, and phenotypic diversity, more systematic guidance on hyperparameter selection (or robustness
  analysis across a wider range) would be valuable for practitioners.
  4. Scalability discussion is limited. The paper argues that the contrastive formulation
  avoids the O(C) summation of softmax by summing over minibatch size M instead. However, the class vectors
  W ∈ R^{D×C} still grow with C, and the attractive update requires indexing into all C columns. For very
  large screens (C ~ 10^6 as mentioned), the memory and I/O cost of maintaining and updating W is not
  discussed. Does SPC scale gracefully to the regime the paper motivates?
  5. Presentation of the fine-tuning results could be stronger. Section 3.2.1 shows that SPC
  outperforms WSL as a LoRA fine-tuning objective for DINOv2, but only on one dataset split (cell holdout
  JUMP). The gains of ViT/L over ResNet18 are described as "small," which somewhat undermines the
  fine-tuning narrative. A more thorough comparison across datasets and splits would better support the
  claim that SPC is an effective fine-tuning technique.

---

> ### Author Rebuttal · Authors · 2026-03-30
>
> We thank the reviewer for their detailed and helpful feedback, and address the main comments below.
>
> ## Training dynamics and possibility of collapse
>
> Since multiple reviewers have raised this point, we have collected our reply in the response to reviewer ghvi, which includes both a semi-formal analysis and empirical cosine similarities during training.
>
> ## WSL ResNet18 baseline and comparability with Phenom-1/Phenom-2
>
> Apologies, this may have been unclear. All results on table 8 are with a ResNet18 encoder. We will update the caption for table 8 to specify that a ResNet18 is used. On comparability to private models Phenom-1/Phenom-2, we would argue that RxRx3-core was developed as a benchmark dataset already pre-processed and with evaluation code provided, so that the comparison is reasonably well controlled. The results of OpenPhenom (publicly available) can also be reproduced, giving confidence that the evaluation approach is the same as that used in the original RxRx3-core paper.
>
> ## Ablations and batch size
>
> We give results training with alternative batch sizes on BBBC021. Other examples of varied batch size can be found in the transformer fine-tuning (256) and RxRx3-core training (512).
>
> (BBBC021 table 1 columns)
>
> WSL b=128 | 0.7582 | 0.8508 | 0.8932 | 0.8913
>
> SPC b=128 | 0.8301 | 0.8922 | 0.9126 | 0.9239
>
> WSL b=256 | 0.7385 | 0.8224 | 0.8737 | 0.8695
>
> SPC b=256 | 0.8104 | 0.8691 | 0.8737 | 0.8913
>
> We also provide ablations of the temperature on the HaCaT JUMP dataset. Under these ablations SPC continues to outperform WSL on 10/12 metrics including all mAP scores which measures broader phenotypic grouping.
>
> (HaCaT table 3 columns)
>
> SPC t=0.01 | 0.492 | 0.193 | 0.397 | 0.177 | 0.520 | 0.277
>
> SPC t=0.25 | 0.476 | 0.196 | 0.476 | 0.196 | 0.467 | 0.258
>
> ## Scalability
>
> SPC updates prototypes only for classes that were seen in the minibatch; since indexing into a matrix is O(1) this makes the update step O(M) where M is the batch size. It is true that there is a storage requirement as the size of W will grow with the number of classes, however the classes would have relatively few samples compared to the size of the dataset so they could be split such that any one node has much fewer than C vectors. On clock speed, we would not expect any speed improvements at the small class numbers used, where the batch size is the same order of magnitude as the classes. We mention it as a positive factor for future scaling.
>
> ## Limited reporting of fine-tuning
>
> We expand on the HaCaT JUMP fine-tuning reported in the paper by providing results for the HaCaT CRISPR cell holdout dataset.
>
> Model | 1-NN | mAP
>
> WSL DinoV2 base | .635 | .288
>
> SPC DinoV2 base | .744 | .460
>
> SPC again outperforms WSL, and in this case the dino base version also outperforms the fully trained ResNet18 on table 3. SPC DinoV2 large did not finish training in time for the rebuttal, but as of writing is at epoch 89 with .767/.512, a more substantial improvement over ResNet18. We will include these results with the completed dino large in the final version.

---

> > ### Author Rebuttal · Reviewer_qP7b · 2026-04-05
> >
> > The authors have addressed my main concerns satisfactorily.  I maintain my score of 4.

---

### Official Review · Reviewer_ZxCm · 2026-03-13

**Soundness:** 3
**Presentation:** 2
**Significance:** 3
**Originality:** 2
**Overall Recommendation:** 4
**Confidence:** 3

**Summary:**

The paper proposes Spherical Phenotype Clustering (SPC) for grouping phenotypically similar perturbations in high-content screening. The key idea is to use experimental metadata as class prototype vectors on a unit hypersphere, and to train these prototypes with attraction-only updates while the image encoder still sees both attractive and repulsive terms. The motivation is that standard contrastive losses would push treatments with similar biology apart, whereas dropping repulsion from the prototype update lets similar classes naturally overlap. Results are shown on BBBC021, a new HaCaT cell screen the authors ran themselves, and RxRx3-core, where SPC with an 11M ResNet18 beats Phenom-1 (307M parameters) on several metrics.

**Compliance With Llm Reviewing Policy:**

Affirmed.

**Final Justification:**

Soundness is good, the rebuttal addressed the collapse and initialization concerns well, and the evaluation across three datasets including uncurated wet-lab data is thorough. Presentation has issues but these are revision-addressable per the authors' promised changes. What keeps this at a weak accept is originality and significance: the contribution is a domain-specific combination of known components, and the benefit of the asymmetric update is most pronounced for NTXent with less consistent gains for SupContrast. The efficiency story is compelling but the foundation model comparisons need the more careful scoping the authors have agreed to. The rebuttal resolved my main technical concerns but reinforced rather than changed my overall assessment, so I'm maintaining my score.

**Key Questions For Authors:**

1. Can you provide a direct analysis of prototype vector behavior during training, such as pairwise cosine distances between prototypes, to support the claim that the no-repulse update does not lead to collapse?
2. How sensitive is SPC to the initialization of the class vectors W, and how does optimization behave in the early stages of training given the small batch size relative to the number of classes?

**Limitations:**

yes

**Strengths And Weaknesses:**

Solid paper with a clear practical motivation and honest evaluation including real wet-lab data. The asymmetric update is a neat idea and the efficiency gains over large foundation models are genuinely useful. Main concerns are on the presentation side, the method definition comes too late and the methods section is cluttered with background material, and there are gaps around prototype dynamics that the paper doesn't fully address. The individual components of SPC are all prior art and the contribution is a domain-specific combination, though a well-motivated one. Claims about outperforming large models also need to be scoped more carefully given Phenom-2 results.

**Soundness**

Strengths:

1. Diverse benchmarking and wet lab validation: The method is evaluated on three datasets, including wet-lab experiments the authors ran themselves. The inclusion of an uncurated real-world screen adds genuine credibility to the claims.
2. Mathematical grounding: The authors provide a geometric derivation in Appendix A connecting the gradient of the cosine similarity loss to the tangent of the spherical arc, which provides a principled motivation for the attraction-only update.

Weaknesses:

3. Prototype update noise and initialization sensitivity: The prototype updates are noisy by design, batch size 16 with many classes likely means one sample per class per update. The paper doesn't discuss this at all, and relatedly doesn't discuss initialization of W or sensitivity to it. Bad initialization plus noisy updates is a combination that could matter and deserves at least some analysis.
4. Lack of collapse-prevention analysis: It's not theoretically obvious why dropping repulsion between prototypes doesn't lead to collapse. The paper's answer is essentially "the encoder still has repulsion so it's fine," but this remains an informal argument that relies on empirical observation rather than any theoretical backing. The t-SNE plots (Figures 6b, 8) suggest collapse isn't happening in practice since you can see meaningful cluster structure, but they show image embeddings not the prototype vectors directly. Showing pairwise prototype distances during training would make this argument more rigorous.
5. Limited analysis of the no-repulse update for SupContrast: The no-repulse update only consistently helps NTXent, not SupContrast (NSC/NSCB barely move for SupContrast w/o repulse in Table 1). The paper notes this but doesn't explain it, which feels like a missed opportunity.

**Presentation**

Strengths:

6. Clear motivation: The problem being addressed is well-chosen and the limitations of standard contrastive learning in this setting are clearly identified, even if the overall presentation needs work.

Weaknesses:

7. Structure and clarity: Section 2 reads like a related work section. The actual novel contribution is buried at the end after pages of background on WSL, NTXent and SupContrast.
8. Delayed method definition: You don't learn that SPC = "NTXent w/o repulse" until Section 3.1 when the ablation table appears. This should be stated upfront in the methods.
9. Missing algorithm block: No pseudocode or algorithm block. For a method with alternating updates and two separate optimizers, this is a real reproducibility issue.

**Significance**

Strengths:

10. Compute efficiency: The efficiency story is compelling. Beating Phenom-1 with 25x fewer parameters on a realistic benchmark is a result practitioners will care about.

Weaknesses:

11. Scale limitations: Phenom-2 still wins on CORUM (.486 vs .428) and HuMAP (.553 vs .498), and SPC also narrowly loses to Phenom-1 on Z-score (4.31 vs 4.35). The narrative around outperforming large foundation models should be more careful about where this holds and where it doesn't.

**Originality**

Strengths:

12. Asymmetric update mechanism: The asymmetric update is a simple and effective idea that fits the biological screening setting naturally.

Weaknesses:

13. Incremental combination: The individual components, NTXent, hyperspherical embedding, and geodesic prototype updates, are all established in prior work. The contribution is a well-motivated domain-specific combination rather than something fundamentally new.
14. Loss-function dependence: The asymmetric update benefit being specific to NTXent limits how broadly applicable the technique appears.

---

> ### Author Rebuttal · Authors · 2026-03-30
>
> We thank the reviewer for their praise and detailed feedback, and give our responses below.
>
> ## Possibility of collapse, noisiness and effects of initialization
>
> Since multiple reviewers have asked about this topic, we have collected our reply in the response to reviewer ghvi. We include both a semi-formal discussion and empirical cosine similarities during training. To answer the initialization question we also show the result of initializing vectors in the collapsed state, where the prototypes escape and show inter-prototype distances similar to a random initialization. Regarding noisiness during training, we show here performance for SPC over the first 5 epochs for 3 training runs, giving mean ± s.d. for evaluation mAP (well profile) each epoch.
>
> SPC | 0.666 ± 0.008 | 0.711 ± 0.008 | 0.731 ± 0.014 | 0.752 ± 0.008 | 0.759 ± 0.007
>
> There is reasonably limited difference across the random initializations. Figure 3 in the article also shows performance during training.
>
> ## Presentation and inclusion of pseudocode
>
> Our thinking on starting with WSL and introducing SPC afterwards is that readers may be familiar with WSL and that since the methods have some similarities it provides a good foundation for understanding how our approach differs. However, we appreciate that the “SPC = NTXent w/o repulse” terminology may seem abrupt in section 3.1 and will state in the methods section when introducing SPC that we elsewhere refer to it as NTXent w/o repulse. On the narrative around outperforming foundation models needing more qualification in places, to the sentence on line 367 that begins “We find higher performance than Phenom-1” we will add “except on the Z-score”. In the final sentence of the introduction, rather than "100x" we will state “much higher parameter counts”. In the conclusion we will change “or outperform” to “or in some cases outperform”. With regards to point 9 about pseudocode, we will include an algorithm pseudocode block in the appendix.
>
> ## Broader applicability of method
>
> Although our asymmetric update is designed with NTXent, there may be potential for such asymmetric updates to be used in other settings such as multi-modality or with alternative loss formulations. The method as it stands may be useful in domains where there are similar hierarchical categories, perhaps something like facial recognition or plant identification.

---

> > ### Author Rebuttal · Reviewer_ZxCm · 2026-04-04
> >
> > Thanks for the response. The semi-formal collapse argument combined with the empirical inter-prototype similarity data and collapsed initialization experiment is convincing. The intra-MoA prototype similarity comparison also directly supports the core claim. The promised text revisions adequately address the remaining presentation concerns. I am maintaining my score.

---

### Official Review · Reviewer_ghvi · 2026-03-14

**Soundness:** 3
**Presentation:** 3
**Significance:** 3
**Originality:** 3
**Overall Recommendation:** 4
**Confidence:** 4

**Summary:**

This paper proposes SPC for learning image representations in HCS. The key idea is to inject treatment metadata into contrastive losses via learnable class vectors, then asymmetrically update these vectors using only the attractive term along spherical geodesics, while the encoder retains the full contrastive loss with repulsive terms. The authors prove a geometric correspondence between the cosine similarity gradient and the geodesic tangent. Experiments show that SPC with ResNet18 outperforms or matches methods with more parameters.

**Compliance With Llm Reviewing Policy:**

Affirmed.

**Key Questions For Authors:**

Can you provide a formal or semi-formal argument for why encoder-side repulsion suffices to prevent collapse when class vectors are updated with attraction only? The ablation shows it works empirically, but understanding the conditions under which this guarantee holds (or might break) would significantly strengthen the contribution.

**Limitations:**

yes

**Strengths And Weaknesses:**

## Strengths

- Simple yet highly effective. The asymmetric design—removing repulsive terms only from class vector updates—is elegant and easy to implement.

-  The evaluation is broad, covering three datasets with diverse metrics. Although the contribution may be considered somewhat marginal for a top venue, I personally find the approach elegant and the results convincing.

## Weakness
- Nearly all experiments rely on ResNet18. The only ViT experiment uses LoRA fine-tuning on a single dataset (HaCaT-CRISPR).  It is unclear whether SPC's advantages hold with other architectures such as fully-trained vision transformers.

---

> ### Author Rebuttal · Authors · 2026-03-30
>
> We thank the reviewer for praising the elegance of the approach and the suggestion to pursue further analysis of the training dynamics.
>
> ## Encoder only repulsion: semi-formal argument
>
> Consider loss function (2) in the main article with logarithm expanded
>
> $$
> L = \sum_{i=1}^M \left( - \log \textnormal{sim}(z_i, W_{y_i}) + \log \sum_{j = 1}^M{[\textnormal{sim}(z_i, W_{y_j}) + \textnormal{sim}(z_i, z_{j \ne i})]} \right)
> $$
>
> We provide a toy analysis with a batch size of two, so that we have two samples $z_1$ and $z_2$ with associated prototypes $W_{y_1}$ and $W_{y_2}$. Writing out terms explicitly gives
>
> $$
> L = - \log \textnormal{sim}(z_1, W_{y_1}) + \log [\textnormal{sim}(z_1, W_{y_1}) + \textnormal{sim}(z_1, W_{y_2}) + \textnormal{sim}(z_1, z_2)] - \log \textnormal{sim}(z_2, W_{y_2}) + \log [\textnormal{sim}(z_2, W_{y_1}) + \textnormal{sim}(z_2, W_{y_2}) + \textnormal{sim}(z_2, z_1)]
> $$
>
> Assume all prototypes exist at a point and the encoder is in an entirely collapsed state so that similarities are identical $s = \textnormal{sim}(z_i, W_{y_j}) = \textnormal{sim}(z_1, z_2)$. Then we have
>
> $$
> L_{collapse} = - \log s + \log [s + s + s] - \log s + \log [s + s + s] =  2 \log 3
> $$
>
> Now consider a mild perturbation of the encoder, so that $z_1$ and $z_2$ no longer sit exactly on top of the prototype vectors $W$. Assume for simplicity that they deviate the same distance from the collapsed $W$ so that we now have $\textnormal{sim}(z_1, W_{y_1}) = \textnormal{sim}(z_2, W_{y_2}) = \alpha s$ where $\alpha$ is some scalar value. We also have $\textnormal{sim}(z_1, z_2) = \beta s$ where $\beta$ is another scalar. Then the new loss function becomes
>
> $$
> L_{perturb} = - \log \alpha s + \log [2 \alpha s + \beta s] - \log \alpha s + \log [2 \alpha s + \beta s] =  2 \log[ 2 + \frac{\beta}{\alpha} ]
> $$
>
> Then as $\beta \rightarrow 0$ we have $L_{perturb} \rightarrow 2 \log 2 < 2 \log 3 = L_{collapse}$ and more generally as long as $\beta < \alpha$ we have that $L_{perturb} < L_{collapse}$. Remembering $s$ is a similarity, this implies that as long as the distance between $z_1$ and $z_2$ is greater than their distance to the prototypes, the perturbed loss function is smaller than at the collapsed encoder state. Since the perturbed samples can be made arbitrarily close to $W$ while maintaining this property, there must be local directions of the loss function smaller than the collapsed state, implying the existence of a gradient in descent directions. The prototype update will then move the $W_{y_i}$ toward the now separated $z_i$ inducing a separation in the previously collapsed $W$.
>
> This analysis indicates that collapse is not a stable state for either the encoder or the prototypes, and that in such a state slight perturbations of the encoder lead to reduced loss and separated samples and prototypes. More generally because the prototype update moves prototypes towards the class-conditional batch mean, the prototypes would only collapse globally if the encoder collapsed first, but the preceding analysis suggests this is unlikely to occur.
>
> With larger batch size, assuming $M < D$ and that all inter-sample similarities $\beta s$ are equal (for example sitting on the vertices of a regular simplex), then an almost identical analysis shows $L_{perturb} = M \log[ M + (M-1) \beta/\alpha]$ with $\beta = \alpha$ in the collapsed state, leading to the same conclusions as in the case with a batch size of two.
>
> Note the importance of the instance-instance terms in providing a de-collapsing gradient. Without them, since the $W$ are all equal any perturbation of the encoder gives
>
> $$
> L = \sum_{i=1}^M \left( - \log \alpha_i s + \log \sum^M_{j=1} \alpha_i s \right) = M \log M
> $$
>
> So the encoder would not escape collapse.
>
> ## Encoder only repulsion: empirical cosine similarities
>
> We provide mean inter-prototype similarities during training, as well as inter-prototype similarities calculated only between prototypes of the same moa (intra-moa). We do this on BBBC021 which has very reliable moa annotations. The results show empirically SPC does not collapse (compare cosine loss), and even with a collapsed initialization escapes to train successfully. They also show that SPC prototypes of the same moa are packed much closer together than those of WSL, supporting the discussion in the article.
>
> Epoch | 0 | 24 | 49 | 74 | 99
>
> Inter-prototype cosine similarities
>
> SPC (random init) | .010 | .069 | .097 | .116 | .120
>
> SPC (collapsed init) | .999 | .068 | .096 | .111 | .134
>
> WSL | .008 | .118 | .121 | .126 | .134
>
> Cosine | .010 | .999 | .999 | .999 | .999
>
> Intra-moa prototype cosine similarities
>
> SPC | .194 | .827 | .829 | .826 | .827
>
> WSL | .194 | .555 | .479 | .455 | .450
>
> NTXent w/ metadata | .194 | .780 | .766 | .757 | .750
>
> We will add the semi-formal argument and graphs of the empirical results to the appendix of the paper.
>
> ## Limited ViT results
>
> We report additional results fine-tuning ViTs in our response to qP7b who also raised this topic.

---

> > ### Author Rebuttal · Reviewer_ghvi · 2026-04-04
> >
> > Thank you for taking the time to provide a thorough explanation. I believe the reasoning is clear and well-supported, and I have no additional questions.

---

### Decision · Program_Chairs · 2026-04-30

**Decision:**

Accept (regular)

**Comment:**

This paper proposes a method to improve representaiton learning for high content screening by modifying the objectives to reflect biological similarity through the concept of an asymmetric contrasive learning process. The reviewers agree that the paper presents a simple and elegant idea with strong empirical validation across multiple benchmarks. There were some concerns about theoretical depth and novelty, but there is a consensus that the paper is technically sound and useful to the community. I recommend weak accept.